

# Analysis of the risk associated to coastal flooding hazards: A new historical extreme storm surges dataset for Dunkirk, France

**Yasser HAMDI[1], Emmanuel GARNIER[2], Nathalie GILOY[1], Claire-Marie. DULUC[1], Vincent REBOUR[1]**

[1] {Institute for Radiological Protection and Nuclear Safety, BP17, 92 262 Fontenay aux Roses Cedex, France}
[2] {UMR 6249 CNRS Chrono-Environnement, Besançon, France}

Correspondence to: Y. Hamdi (yasser.hamdi@irsn.fr)

## Abstract

This paper aims to demonstrate the technical feasibility of a historical study devoted to French Nuclear Power Plants (NPPs) which can be prone to extreme marine flooding events. It has been shown in the literature that the use of HI can significantly improve the probabilistic and statistical modelling of extreme events. There is a significant lack of historical data about marine flooding (storms and storm surges) compared to river flooding events. To address this data scarcity and to improve the estimation of the risk associated to the marine flooding hazards, a dataset of historical storms and storm surges that hit the Nord-Pas-de-Calais region during the five past centuries were recovered from archival sources, examined and used in a frequency analysis (FA) in order to assess its impact on the frequency estimations. This work on the Dunkirk site (representative of the Gravelines NPP) is a continuation of previous work performed on the La Rochelle site in France. Indeed, the frequency model (FM) used in the present paper had some success in the field of coastal hazards and it has been applied in previous studies to surge datasets to prevent marine flooding in the La Rochelle region in France.

In a first step, only information collected from the literature (published reports, journal papers and PhD theses) is considered. Although this first historical dataset has extended the gauged record back in time to 1897, serious questions related to the exhaustiveness of the information and about the validity of the developed FM have remained unanswered. Additional qualitative and quantitative HI were extracted in a second step from many older archival sources. This work has led to the construction of storms and marine flooding sheets summarizing key data on each identified event. The quality control and the cross-validation of the collected information, which have been carried out systematically, indicate that it is valid and complete as regards extreme storms and storm surges. Most of the HI gathered displays a good agreement with other archival sources and documentary climate reconstructions. The probabilistic and statistical analysis of a dataset containing an exceptional observation considered as an outlier (i.e. the 1953 storm surge) has been significantly improved when the additional HI gathered in both literature and archives are used. As the historical data tend to be extreme, the right tail of the distribution has been reinforced and the 1953 "exceptional" event don't appear as an outlier any more. This new dataset provides a valuable source of information on storm surges for future characterization of coastal hazards.

**Key-words:** Coastal storms; Storm surges; Marine flooding; Historical information; Frequency analysis;

## 1 Introduction

As the coastal zone of the Nord-Pas-de-Calais region in Northern France is densely populated, marine flooding represents a natural hazard threatening the costal populations and facilities in several areas along the shore. The Gravelines Nuclear Power Plant (NPP) is one of those coastal facilities. It is located near the community of Gravelines in North France, approximately 20 km from Dunkirk and Calais. The Gravelines NPP is the sixth largest nuclear power station in the world, the second largest in Europe and the largest in Western Europe.

Extreme weather conditions could induce strong surges that could cause marine submersion. The 1953 North Sea flood was a major flood caused by a heavy storm that occurred on the night of Saturday, 31 January and morning of Sunday, 1 February. The floods struck many European countries and France had not been the exception. This was particularly the case along the northern coast of France, from Dunkirk to the Belgium border. The site of Dunkirk is the site of interest in the present paper (Fig. 1 to the left). An old plan of the Dunkirk city is presented in the right panel of Fig. 1 (we shall return to this plan at a later stage in this paper). It's a common belief today that the Dunkirk region is vulnerable and subject to several climate risks (e.g., Maspataud et al. 2013). More severe marine flooding events such as the November 2007 North Sea and the March 2008 Atlantic storms, could have had much more severe consequences especially if they



occurred at high tide (Maspataud et al. 2013; Idier et al. 2012). It is then important for us to take into account
the return periods of such events (especially in the current context of global change and projected sea-level
rise) in order to manage and reduce coastal hazards, implement risk policies prevention and to enhance and
strengthen coastal defence against marine flooding.
The storm surge frequency analysis (FA) represents a key step in the evaluation of the risk associated to
coastal hazards. The frequency estimation of extreme events (induced by natural hazards) using probability
functions has been extensively studied for more than a century (e.g., Gumbel, 1935; Chow, 1953; Dalrymple,
1960; Hosking and Wallis, 1986, 1993, 1997, Hamdi et al. 2014, 2015). We generally need to estimate the
risk associated to an extreme event of a given return period. Most extreme value models are based on
available at-site recorded observations only. A common problem in FA and estimation of the risk associated
to extreme events is the estimation from a relatively short gauged record of the flood corresponding to 100-
1000 years return periods. The problem is even more complicated when this short record contains an outlier,
(an observation much higher than any other ones in the dataset). This is the case for several sea level time
series in France and unfortunately this characterizes the Dunkirk surge time series as well.
The 1953 storm surge was considered as an outlier in our previous work (Hamdi et al., 2014) and in
previous researches (e.g., Bardet et al., 2011). Indeed, although the Gravelines NPP is designed to very low
probabilities of failure and despite the fact that no damage was reported at the French NPPs, the 1953
marine flooding had shown that the extreme sea levels estimated with the current statistical approaches
could be underestimated. It seems that the local FA is not really suitable for a relatively short dataset
containing an outlier.
Indeed, a poor estimation of the distribution parameters may be related to the presence of an outlier in
the sample (Hamdi et al., 2015), they must be properly addressed in the FA. One would expect that one or
more additional extreme events in a long period (500 years for instance) would, if properly included in the
frequency model (FM), improve the estimation of a quantile at the given high return period. The use of other
sources of information with more adapted FMs is required in the frequency estimation of extremes. Worth
noting is that this recommendation is not new and dates from several years. The value of using other
sources of data in the FA of extreme events has been recognized by several authors (e.g. Hosking and
Wallis, 1986 and Stedinger and Cohn, 1986). By other sources of information we refer here to events
occurred not only before the systematic period (gauging period) but also during gaps of the recorded time
series. Water marks left by extreme floods, damage reports and newspapers are reliable sources of
Historical information (HI). It can also be found in the literature, archives and unpublished written records,
etc. It may also arise from verbal communications from the general public. Paleoflood and dendrohydrology
records (the analysis and application of tree-ring records) can be useful as well. A literature review on the
use of HI in flood FAs with an inventory of methods for its modeling has been published by Ouarda et al.,
(1998). Attempts to evaluate the usefulness of the HI for the frequency estimation of extreme events are
numerous in the literature (e.g. Guo and Cunnane, 1991; Ouarda et al. 1998; Gaal et al., 2010; Payrastre et
al., 2011; Hamdi, 2011; Hamdi et al. 2015). Hosking and Wallis (1986) have assessed the value of HI using
simulated flood series and historical events generated from an extreme value distribution and quantiles are
estimated by the maximum likelihood method with and without the historical event. The accuracy of the
quantile estimates was then assessed and it was concluded that HI is of great value provided either that the
flood frequency distribution has at least three unknown parameters or if gauged records are short. It was also
included that the inclusion of HI is unlikely to be useful in practice when a large number of sites are used in a
regional context. Because HI is often imprecise, their inaccuracy should be considered in the analysis.
Nevertheless, the influence of an outlier can be decreased by increasing its representativity in the sample
when using the HI, knowing that its uncertainty is sometimes important (e.g. Payrastre et al. 2011; Hamdi et
al. 2015). A frequency estimation of extreme storm surges based on the use of HI has rarely been explicitly
studied in the literature (Bulteau et al., 2014, Hamdi et al. 2015, 2016) despite its significant impact on social
and economic activities and on NPPs' safety. Bulteau et al. (2014) have estimated extreme sea-levels by
applying a Bayesian model to the La Rochelle site in France. This same site was used as a case study by
Hamdi et al., (2015) to characterize the marine flooding hazard. The use of a skew surge series containing
an outlier in local frequency estimation is limited in the literature as well.
It is often possible to augment the storm surges record with those occurred before and after gauging
began. Before embarking on a thorough and exhaustive research of any HI related to coastal flooding that hit
the area of interest, potential sources of historical marine flooding data for the French coast (Atlantic and
English Channel) and more specifically for the Charente-Maritime region were identified in the literature (e.g.
Garnier and Surville, 2010). The HI gathered has been very helpful in the estimation of extreme surges at La
Rochelle which was heavily affected by the storm Xynthia in 2010 that generated a water level considered so
far as an outlier (Hamdi et al., 2015). Indeed, these results for the La Rochelle site have encourage us to
build a more complete historical database covering all the extreme marine flooding occurred over the five
past centuries in the entire French coast (Atlantic and English Channel). However, only the historical storm
surges that hit the Nord-Pas-de-Calais region during this period are presented herein.





The main objective of the present work is collect HI about storms and storm surges occurred in the last five centuries and to examine its impact on the frequency estimation of extreme storm surges. The paper is organized as follows. HI gathered in the literature and its impact on the FA results is presented in sections 2 and 3. The fourth section presents the HI recovered from archival sources, their quality control and validation. In the section 5, the FM is applied using both literature and archival sources. The results are discussed in the same section before concluding and presenting some perspectives in section 6.

## 2  How HI improve effective design?

The effective design of the coastal defense is dependent on how high a design quantile (1000-years storm surge for instance) will be. But this is always estimated with uncertainty and not precisely known. Indeed, any frequency estimation is given with a confidence interval (CI) whose width depends mainly on the size of the sample used in the estimation. Some other sources of uncertainties (such as the use of trends related to the climatic evolution) can be considered in the frequency estimation. As mentioned in the introductory section, samples are often short and characterized by the presence of outliers. The CIs are rather large and in some cases exceed 2 or 3 times (and even more) the value of the quantile. Using the upper limit of this CI would likely lead to more expensive design of the defense structure. One could just use the most likely estimate and neglect the CI but it is more interesting to consider the uncertainty as often estimated by the probabilistic engineers. The width of the CI (i.e. inversely related to the sample size) can be reduced by increasing the sample size. In the present work, we focus on increasing the number of observations by adding information about storm surges induced by historical events. Storm events for which surges are available can be subdivided into three groups, on the base of surge data availability:
1. Systematic records at tide gauges
2. Short-term HI (extracted from the literature)
3. Medium and long-term pregauged HI (can be found in archives and gathered by historians)

### *2.1  Systematic surge records*

The surge dataset is obtained from the corrected observations and predicted tide levels. The tide gauge data is managed by the French Oceanographic Service (SHOM - Service Hydrographique et Océanographique de la Marine) and measures are available since 1956. The R package TideHarmonics (Stephenson, 2015) is used to calculate the tidal predictions. In order to remove the effect of sea level rise, the initial mean sea level (obtained by tidal analysis) is corrected for each year by using an annual linear regression, before calculating the predictions. The regression is obtained by calculating daily means using a Demerliac Filter (Simon 2007). Monthly and annual means are calculated respecting the Permanent Service for Mean Sea Level (PSMSL) criteria (Holgate, et al., 2013). This method is inspired by the method used by SHOM for its analysis on high water levels during extreme events (SHOM, 2015). The available systematic surge dataset was obtained for the period from 1956 to 2015.

### *2.2  Short-term HI*

#### *2.2.1  HI during gaps of systematic records*

As mentioned above, a common issue in frequency estimation exercises is the presence of gaps within the dataset. Failure of the measuring devices and damages (whose main cause is especially the natural hazards) are often the origin of these gaps. They may also be due to human errors, strikes, wars, etc. Nevertheless, failures in measuring stations (occurred during a storm for instance) creating gaps are themselves non-independent events. It is therefore necessary to ensure that the occurrence of the gaps and the observed variable are independent. Whatever the origin and characteristics of the missing period, the use of the full set of the extreme storm surges occurred during the gaps is strongly recommended to ensure the exhaustiveness of the information. This will make the estimates more robust and reduce associated uncertainties. Indeed, by delving into the literature and the web, one can obtain more information about this kind of events. Maspataud (2011) was able to gather sea level measurements that were taken by regional maritime services during a storm event in the beginning of 1995, a time where the Dunkerque tide gauge was not working. This allowed the calculation of the skew surge, which was estimated by the author at 1.15 m on January $2^{nd}$, 1995. This storm surge is high enough to be considered extreme. In fact, it was exceeded only twice during the systematic period (January $5^{th}$, 2012 and December $6^{th}$, 2013). For convenience, we would like to recall here the definition of a skew surge: It is the difference between the maximum observed water level and the maximum predicted tidal one regardless of their timing during the tidal cycle (a tidal cycle contains one skew surge).



*2.2.2 Short-term pregauged HI*
A literature review was conducted in order to get an overview of the storm events and associated surges that
hit the Nord-Pas-de-Calais region in France during the last two centuries. Some documents and storm
databases on local, regional or national scales are available:
• the " Plan de Prévention de Risques Littoraux (PPRL) " : are documents made by the French state on a
communal scale, describing the risks a coastal zone is subject to, e.g. marine inundation and coastal
erosion, preventive measures in case of a hazard happening. To highlight the vulnerability of a zone, an
inventory of storms and marine inundation within the considered area is attached to this document.
• Deboudt (1997) and Maspataud (2011) describe the impact of storms on coastal areas for the study
region;
• the VIMERS Project: gives information on the evolutions of the Atlantic depressions that hit Brittany
(DREAL Bretagne 2015);
• NIVEXT Project: presents historical tide gauge data and the corresponding extreme water and surge
levels for storm events (SHOM, 2015) ;
• Lamb 1991 : provides a synoptic reconstructions of the major storms that hit the British isles for the 16th
century up to today

According to the literature, the storm of the 31st January to 1st February 1953 caused the greatest surge
and was the most damaging within the study area. This event is well analyzed and documented (Sneyers,
1953, Rossiter 1954, Gerritsen, 2005, Wolf and Flather 2005): A depression formed over the Northern
Atlantic Ocean close to Iceland moving eastward over Scotland and then changing its direction to south-
eastwards over the North Sea was accompanied by strong northerly winds. An important surge was
generated by this storm that, in conjunction with a high spring tide, resulted in particularly high sea levels.
Around the southern parts of the Northern Sea the maximum surges exceeded 2.25 m reaching 3.90 m at
Harlingen, Netherlands, large areas were flooded in the Great Britain, Northern Parts of France, Belgium, the
Netherlands and the German Bight, causing the death of more than 2000 people. Le Gorgeu and
Guittonneau (1954) indicate that during this event, the water level exceeded over 2.40 m the predicted water
level at the Eastern Dyke of Dunkirk. Bardet et al. (2011) included a storm surge equal to 2.13 m in the
developed regional model. Both authors indicate the same observed water level, i.e. 7.90 m but the
predicted water level differs: While in 1954 the predicted water level was estimated at 5,50 m, the predictions
were reevaluated to 5.77 m by the SHOM using the harmonic method. A storm surge of 2.13m is therefor
used in the present study. Nevertheless, some other storms causing important surges and flooding occurred
within the area of interest, these events are listed in the Table 1. Three of these storms are quite well
documented within the literature and are illustrated below:
**14/01/1808:** During the night from 14th to 15th January 1808, "a terrible storm, similar to a storm that hit the
region less than one year before on 18 February 1807" hit the coasts of the most northern parts of France up
to the Netherlands This storm caused severe flooding as well in the Dunkirk area as also in Zeeland area in
the south western parts of the Netherlands where the water rose up to 25 feet on the isle of Walcheren (i.e.
7,62 m). The journal also reports more than 200 deaths. For the Dunkirk area, the last time the water levels
rose as high as in January 1808 was 2nd February 1791. Unfortunately this source did not provide any
information that we can quantify or any information on the meteorological and weather conditions we can use
to reconstruct the storm surge value.
**28/11/1897:** What was felt as stormy winds were felt in Ireland on the 27th November 1897 became an
eastward moving storm with gale force winds over Great Britain, Denmark and Norway (Lamb, 1991). This
storm caused interruption of telephone communications between the cities of Calais, Dunkirk and Lille and
great damage to the coastal areas (Le Stéphanois, November 30th 1897). At Malo-les-Bains, a small town
close to Dunkirk, the highest water level reached 7.36 m although the high tide was predicted with 5.50 m,
resulting in a skew surge of 1.86 m that caused huge damage to the port infrastructures (DREAL Nord – Pas
de Calais)
**01/03/1949:** A violent storm with mean hourly wind speeds reaching almost 30 m.s$^{-1}$ and gusts up to 38.5
m.s$^{-1}$ (Volker, 1953) was the cause of a storm surge that reached coast of the North of France and Belgium
in the beginning of March 1949. The tide gauge of Antwerp in the Escaut estuary measured a water level
higher than 7 m TAW which classifies this event as a "*buitengewone Stormvloed*", an extraordinary storm
surge (Codde and De Keyser 1967). For Dunkirk area two sources reporting water levels were found: The
first saying that 7.30 m was reached as a maximum water level at the eastern Dike in Dunkirk, exceeding the
predicted high tide, i.e. 5.70 m, with 1.60 m (Le Gorgeu and Guittoneau 1954). A second document relates
that the maximum reached was about 7.55 m at Malo-les-Bains, which would mean a surge of 1.85 m
(DREAL Nord – Pas de Calais).
It is worth noting that the use of proxy data (i.e. the descriptions of events in the historical sources
summarized in Table 1) to extract sea level values and to create surges database is seriously limited. For the



1791 and 1808 storms, there is sufficient evidence that extreme surge events have taken place (extreme
water level on Walchern Island) but the sources are not informative enough to estimate water levels reached
in Dunkirk. A surge of 1.25 m is given for the storm of 1921. The problem is that the type of the surge
(instantaneous or skew), the exact location at which it was recorded and the hydro-meteorological
parameters are not informed. For the skew surge of 1949, two different values at two locations are given.
There are predicted and observed water levels for the storms of 1905 and 1953 in Calais, which indicated
that the differences is a skew surge, but likewise neither the exact location nor the information about the
reference level were furnished. The need of tracing back to "direct data" describing a storm and its
consequences becomes clear, as well as performing a cross-check of the data on a spatial and factual level,
as Brázdil (2000) also suggests. Only the 1897, 1949, 1953 and 1995 events are considered in the present
work (Table 2). Another important question arises, while trying to inventory events, is related to the
exhaustiveness of the HI gathered in a well-defined time-window (called hereafter the historical period). In
order to properly perform the FA, this criterion must be fulfilled. Indeed, we are fairly trustful and have good
evidence to believe that other than the 1995 storm surge, the surges induced by the 1897, 1949 and 1953
storms are the biggest on the period 1897-2015.

### 2.3  Long-term pregauged HI

A historical research devoted to the French NPPs located at the Atlantic and English Channel coast is a
genuine scientific challenge due to the timely implementation and the geographic dispersion of the nuclear
sites. The process involves the exploration and consultation of a large number of historical sources in a
context of a permanent multi-scalar approach. Indeed, NPPs are generally implemented, for obvious safety
reasons, in sparsely populated and isolated areas. Ransom that choice, these sites knew little anthropogenic
influence in the past. However, this difficulty does not mortgage a historical perspective due to the rich
documentary resources for studying an extreme event to different scales ranging from the site itself to that of
the Region (Garnier, 2015 and 2017 bis). In addition, this may be an opportunity for researchers and a part
of the solution because it also allows a risk assessment at ungauged sites.

## 3  Extreme storm surge frequency estimation using systematic records and short term HI

In this work, we suggest a method of incorporating the HI developed by Hamdi et al. (2015). The proposed
FM (POTH) is based on the Peaks-Over-Threshold with HI. The POTH method uses two types of HI: Over-
Threshold Supplementary (OTS) and Historical Maxima (HMax) data which are structured in historical
periods. Both kinds of historical data can only be complementary to the main systematic sample. The POTH
FM was applied to the Dunkirk site to assess the value of historical data in marine flooding FA and more
particularly in improving the frequency estimation of extreme storm surges

### 3.1  Settings of the POT frequency model

To prepare the systematic POT sample and in order to exploit all available data separated by gaps, the
surges recorded since 1956 were concatenated to form one systematic series. A POT threshold equal to
0,75 m (corresponding to an events rate equal to 1,4 events/year) is an adequate choice (details about the
threshold selection are not presented herein). The POT sample with an effective duration $w_s$ of 46,5 years
(from 1956 to 2015) is represented by the grey bars in Fig. 2-a and 2-b. As homogeneity, stationarity and
randomness of time series are prerequisites in a FA (Rao & Hamed, 2001), non-parametric tests such as the
Wilcoxon test for homogeneity (Wilcoxon, 1945), the Kendall test for stationarity (Mann, 1945), and the Wald-
Wolfowitz test for randomness (Wald & Wolfowitz, 1943) are applied. These tests were passed by the
Dunkirk station at the 5% level of significance.

### 3.2  Settings of the frequency model with HI (POTH)

The POTH FM was first applied with a single historical data which is that of 1953 represented by the red bar
in Fig. 2-a. It has not been complicated to demonstrate that this event is undoubtedly an outlier. Indeed, in
order to detect outliers, the Grubbs-Beck test was used (Grubbs and Beck, 1972). The reader can be
referred to for more details on this test. As mentioned in the previous section, some historical extreme events
experienced by the Dunkirk city are available in the literature. Only this information (including the 1953 one)
is considered in this first part of the case study.
Otherwise, HI is most often considered in the FA models for pre-gauging data. Less or no attention has
been given to the non-recorded extreme events occurred during the systematic missing periods. As
mentioned earlier in this paper, the sea level measurement induced by the 1995 storm was missed and a
value of the skew surge (1.15 m) was reconstructed from information found in the literature. As this event is





of ordinary intensity and has taken place very recently, it is considered as a systematic data even if this type
of data can be managed by the POTH FM by considering them as HI (Hamdi et al. 2015). The HI gathered is
resumed in Table 2 and the POTH sample with a historical period of 72,51 years is presented in Fig. 3-b.
Parameters characterizing datasets including both systematic and HI were introduced in Hamdi et al., (2015).
The HI is used herein as HMax data that complements the systematic record (with an effective duration $D_{eff}$
equal to $w_s$) on one historical period (1897-2015) with a known duration $w_h = w_{HMax} = 2015 - 1897 + 1 - D_{eff}$ (
$w_h = 72,51 \, years$) and three historical data ($n_k = 3$). Other features of the POTH FM have been used. A
parametric method (based on the Maximum Likelihood) for estimating the General Pareto Distribution (GPD)
parameters considering both systematic and historical data have been developed and used.
The maximum likelihood method was selected for its statistical features especially for large series and for
the ease with which any additional information (i.e. the HI) is incorporated in it. On the other hand, the
plotting positions exceedance formula based on both systematic observations and HI (Hirsch, 1987; Hirsch
and Stedinger, 1987; Guo, 1990) is proposed to calculate the observed probabilities and it has been
incorporated into the POTH FM considered herein. The reader is referred to Hamdi et al. (2015) for more
theoretical details on the POTH model and on the Renext package used to perform all the estimations and
fits.

## 3.3  Results and discussion

We report herein the results of the FA applied to the Dunkirk tide gauge. As with any sensitive facility, high
Return Levels (RLs) (100, 500 and 1000-year extreme surges, for instance) are needed for the safety of
NPPs. The results are presented in form of probability plots in Fig. 2-c&d. The theoretical distribution function
is represented by the solid line in the figures, while the dashed lines represent the limits of the 70% CIs. The
HI is depicted by the empty red circles, while the black full ones represent the systematic sample. The results
(estimates of the desired RLs and uncertainty parameters) are also summarized in Table 3. Fitting the GPD
to the sample of extreme POTH storm surges yields the relative widths $\Delta CI/S_T$ of the 70% CIs (the variance
of the RL estimates are calculated with the delta method).
The FA was firstly performed considering systematic surges and the 1953 storm surge as a historical
data. It can be seen that the fit of the POTH sample including the 1953 historical event (with $w_h$ equal to
16,5 years) presented in Fig. 2-c (called hereafter the initial fitting), is poor at the right tail and more
specifically, at the largest storm surge (the historical data of 2,13 m occurred in 1953) which have a much
lower observed return period than its estimated one. The estimates of the RLs of interest and uncertainty
parameters (the relative width $\Delta CI/S_T$ of the 70% CIs) are presented in columns 2-3 of Table 3. These
initial findings are an important benchmark as we follow the evolution of the results to evaluate the impact of
additional HI. This assessment can also be done by comparing the results HI-using as well as non HI-using.
100, 500 and 1000-year quantiles given by the POTH FM with all the historical data included (called
hereafter the full POTH FM) are about 3- 6% higher than those obtained by the initial POTH FM (this was
expected as the additional historical surges are higher than all the systematic storm surges) and the relative
width of the CIs are about 20-25% narrower.
Unlike the 1897 historical event, the 1949 and 1953 ones has a lower observed return period than their
estimated one. A plausible explanation for this result is that the body of the distribution is better fitted than
the right tail one and this is a shortcoming directly related to the exhaustiveness assumption used in the of
the POTH FM. Indeed, as stated in Hamdi et al. (2015) and as mentioned above, a major limitation of the
developed FM arises when the assumption related to the exhaustiveness of the information is not satisfied.
This is obviously worrying for us because the POTH FM is based on this assumption. Overall using
additional data in the local FM has improved the variances associated to the estimation of the GPD
parameters but did not conduct to robust estimates with a better fitting (particularly at the right tail, the high
RLs being very sensitive to the historical values) if the assumption of exhaustiveness is still strong. This first
conclusion is likewise graphically backed by the CIs plots shown in Fig. 2-d. Nevertheless, as the impact of
historical data becomes more significant, there is an urgent need to carry out a deeper investigation of all the
historical events occurred in the region of interest (Nord-Pas-de-Calais) over a longest historical period. In
order to have robust estimates and reduced uncertainties, it is absolutely necessary that the gathered
information be as complete as possible.

## 4  HI extracted from the archives: a documentary puzzle

To be considered in the FA, a historical storm surge must be well documented; its date must be known and
some information on its magnitude must be available. Mostly, available information concerns the impact and
the societal disruption caused at the time of the event (Baart, 2011).



### 4.1 Historical data sources

First, it is important to distinguish between "direct data" (also referred to as "direct evidence") and "indirect data" (also referred to as "proxy data"). The first refers to all information from the archives that describe an extreme event (a storm surge event for instance) occurred at a known date. If their content is mostly instrumental, such as meteorological records presented in some commonplace books or by the Paris Observatory (since the 17th century), sometimes accurate descriptions of extreme climatic events are likewise found. The "proxy data" rather inform the influence of some storm initiators and triggers such as wind and pressure. Concretely, they provide information indirectly on marine flooding for example.

Private documents or "ego-documents" (accounts and commonplace books, private diaries, etc.) are used in many ways during 16th to 19th Century. Authors recorded local facts, short news and last events, and amongst them, weather incidents. These misidentified historical objects may contain many valuable meteorological data. These private documents most often take the form of a register or a journal in which the authors record various events (economic, social and political) and weather information as well. Other authors used a more integrated approach to describe a weather event by combining observations of extreme events, instrumental information, phenology (impact on harvests), prices in local markets and possibly its social expression (scarcity, emotions, riots, etc.). All these misidentified sources are another opportunity for risk and climate historians to better understand the natural and coastal hazards (marine flooding, earthquakes, tsunamis, landslides, etc.) of the past. Some of these private documents may be limited to weather tables completely disconnected from their socio-economic and climatic contexts. Most of the consulted documents and archives describe the history of marine flooding in the area of interest. Indeed, the historical inventory identifies and describes the damaging marine flooding occurred on the northern coast of France (Nord-Pas-de-Calais and Dunkerque) over the five past centuries. It presents a selection of remarkable marine floods that occurred in this area and it integrates not only the old events but also those occurred after the gauging period has begun. The information is structured around storms and marine submersions summary sheets. Accompanied and supported by a historian, several research and field missions were carried out and a large number of archival sources have been then explored and, whenever possible, exploited. The historical analysis began with the consultation of the documentary information stored in the rich library of the communal archive of Dunkirk, Gravelines, Calais and Saint-Omer. The most consulted documents were obtained directly from the Municipal archives because the Municipal Acts guarantee a chronological continuity at least since the end of the 16$^{th}$ century to the French Revolution (1789). Very useful for spotting extreme events, they unfortunately provide poor instrumental information. Therefor we also considered data out of local chronicles of annals of the city and harbor of Dunkirk, as well as reports written by scientists or naturalists to describe tides at Calais, Gravelines, Dunkirk, Nieuport and Oostende. Most of them contain old maps, technical reports, sketches or plans of dykes, sluices and docks designed by engineers of the 18$^{th}$-20$^{th}$ centuries and from which it may be possible to estimate water levels reached during extreme events. Mostly, the bibliographical documents are chronicles, annals and memoirs written after the disaster. Finally, for the more recent period available local newspapers have been consulted.

Multiplying the sources and trying to crosscheck events allowed us to constitute a database of 81 events. We focused the research on the period between 1500 and 1950, as for most of the time tide gauge observations are available after 1950. The first event took place in 1507 and the last in 1995. Depending on how it is mentioned in the archive and as shown in the left panel of Fig. 3, the collated events were splited in two groups. Storm surge events are events, where there is a clear mention of flooding within the sources. Are considered as storms, events where only information about strong wind and gales are available. Except for 19$^{th}$ century, we have much more storm surge events, than storms events. All the gathered events are summarized in table 4.

### 4.2 Data quality control

All types of data require quality control and need to be corrected and homogenized if necessary to ensure that the data are reflecting real and natural variations of the studied phenomena rather than the influence of other factors. This is particularly the case for historical data that have been taken in different site conditions and have not been taken using modern standards and techniques (Brázdil et al., 2010). As mentioned earlier, archival documents are of different natures and qualities. We therefore decided to classify them by their degree of reliability according to a scale ranging between 1 and 4:

- The degree 1: not very reliable historical source (it is impossible to indicate the exact documentary origin). It is particularly the case for historical information found in the web.
- The degree 2: information found in scientific books talking about storms without clearly mentioning the sources.
- The degree 3: books, newspapers, reports and memories citing historical events and clearly specifying its archival sources.





- The degree 4: is the highest level of reliability. Information is taken in a primary source (e.g., an original archival report talking about a storm written by an engineer in the days following the event).

Although the information classified as a category 1 document is not very reliable, it still gives the information that something happened at a date and is therefore not definitely ignored. Typically this type of document needs to be crosschecked with other documents. As shown in Fig. 3 (to the right), the classification of the data reveals a good reliability of gathered information as there are no sources classified in category 1 and less than 10% of the sources are in category 2. It is worth noting that paradoxically, the older the information, the more reliable the archival document is.

### 4.3 The historical surge dataset

As shown in Section 3, relatively recent events (1897, 1949, 1953 and 1995) have already been quantified and integrated into a FM by assuming that our sources are reliable. It has also been shown that, a database of 75 events (occurred in the period between 1500 and 1950) was constituted. The concern is that it is not always possible to quantify a storm surge or a sea level from the information gathered for each event. We focus herein on the reconstruction of some events of the 18$^{th}$ century where the historical information makes it possible to quantify water levels. Out of the 75 events 40 are identified as events causing an inundation (Fig. 3), but not all the sources contain qualitative data or at least some information about water level reached. We selected herein the events with the most information about some characteristics of the event (the water level reached, wind speed and direction and in some cases measured information). The tide coefficient is a ratio of the semi-diurnal amplitude by the mean spring neap tide amplitude introduced by Laplace in the 19$^{th}$ century and commonly used in France since, then. Today, the coefficient 100 is attributed by definition to the semi-diurnal amplitude of equinox spring tides of Brest. Therefor the range of the coefficient lies between 20 and 120, i.e. the lowest and highest astronomical tides. Calculated for each tide at Brest harbor it is applied to the complete French metropolitan Atlantic and Channel coastal zone (Simon, 2007). Table 5 below shows a synthesis of the six events which we will analyze in more details, showing the tide coefficient we obtained from the SHOM website, wind intensity and direction, a water level reached in Dunkirk, other cities affected and the associated water levels.

**1720-1767:** In essays written by a mathematician of the royal academy of science, De Froucroy D-R, who describes the tide phenomenon in the Flemish coast, extreme water levels observed within the study area are described. During the period 1720 to 1767 the author refers to five events that are confirmed by a Flemish scientist, Dom Mann (1777, 1780). De Froucroy D-R witnessed the water levels induced by the 1763 and 1767 storms and reconstructed the level reached during the 1720 event in Dunkirk. Water levels at that time are given for the cities of Dunkirk, Gravelines and Calais in the "pied du roi" unit (foot of the king was a French measuring unit, corresponding to 0.325 m) above local mean low water springs. The French water levels are completed by measurements made in ancient Flemish feet above highest astronomical tides for the cities of Oostende and Nieuport (De Fourcroy D-R., 1780; Mann, 1777, 1780). Fig. 4 shows an example of HI as presented in the archives (De Fourcroy D-R., 1780).

The 1720 event is a memorable event for the city of Dunkirk, as the spring tide was increased by the strong gales blowing from north-western direction destroyed the cofferdam built by the British in the year 1714, cutting the old harbor from sea access and prohibiting any maritime trade and slowly causing the ruin of the city. The socio-cultural impact of the natural destruction of the cofferdam was huge, as it restarted the trading of the city (Chambre de Commerce de Dunkerque 1895, Plocq, 1873, Belidor, 1788). In 1736, the only sea level available is given for Gravelines harbor, but extreme water levels are confirmed in the sources as they mention at least 4 feet of water in a district of Calais, and water levels that overtopped the docks of the harbor in Dunkirk (Municipal Archive of Dunkirk DK291, Demotier, 1856). As mentioned above, communal and municipal archives contain plans of dykes, dock and sluices of Dunkirk harbor designed by engineers with the means available at that time, and such sketches were recovered. A 1740 sketch showing a profile of the Dunkirk harbor dock is presented in Fig. 5 for illustrative purposes only. The use of these plans and sketches in the quantification of some historical storm surges is ongoing and results will be presented in a future paper. The lower lying streets of Gravelines were accidently flooded by the high water levels in March 1750. The fact that an extreme water level was reported also in Oostende for the same day confirms that the surge was not only a local phenomenon. The surge of 1763 occurred in a period with mean tidal range but water level exceeded the level of mean spring high tide in Dunkirk, Calais and Oostende. Unfortunately no more information about the flooded area is available. Strong west-north-westerly winds caused by a quick drop of the pressure produced high water levels from Calais up to the Flemish cities. It is, at least for the period from 1720 to 1767, the highest water level ever seen and known. The 1720 and 1767 events show good evidence of the wind direction and wind intensity, while, except for the water levels reported, the events from 1736, 1750 and 1763 are in different sources always cited together and described as "*extraordinary sea-levels that are accompanied or caused by strong winds blowing from South-West to North*" (De la Lande, 1781, De Fourcroy D-R., 1780, Mann, 1777, 1780). As with the 1897-2015 historical/systematic periods, the same question related to the exhaustiveness of the HI gathered in the




1720-1770 historical period arises. As our historical research on extreme storm surges occurred in this time-
window was very thorough, we have good reasons to believe that the surges induced by the 1720, 1763 and
1767 storms are the biggest on that historical period.
**1767 – 1897:** For the four events (1778, 1791, 1808 and 1825), the sources report strong winds were
blowing from north-westerly directions and that in Dunkirk the quays and docks of the harbor were
overtopped as the highest water levels were reached. We know that, after the event of February 1825, at
least 19 storms events occurred and we have good evidence to believe that some of them induced extreme
surges, but either the information available is not sufficient to draw an approximate value of the water level.
The quantification of the storm surges induced by these events is complicated and time consuming. To be
able to reduce the CI of the high RLs (the 1000-year one for instance), it is insufficient to have the time-
window (the historical period), as the observations or estimates of high surges are unknown. A fixed time-
window and magnitudes of the available high storm surges are required to improve the estimates of
probabilities of failure. The exhaustiveness assumption of the HI on this time-window will therefore be too
crude and will make no sense. The historical period 1770-1897 was therefore eliminated from inference.
Fortunately, these discontinuities in the historical period have been anticipated in the POTH FM (Hamdi et al.
2015). Two not-successive time-windows 1720-1770 and 1897-2015 will therefore be used as historical
periods in the POTH FM.
**1936:** The 1936 event can be considered as a lower bound, as the document from the archive testifies that
the "water level was at last 1m higher than the predicted tide" during the storm that occurred on the night of
1[st] December 1936 (Municipal Archives of Dunkirk 4S 881). The 1936 event, which can be designated as a
moderately extreme storm, is the only one gathered on the 50-year time-window (1897-1949). As the surge
lower bound value induced by this event is too small (i.e. exceeded more than 10 times during the systematic
period), it could be exceeded several times during the 1897-1949 period. Its involvement in the statistical
inference will have the opposite effect and will not only increase the width of the CI but will also degrade the
quality of the fit. The 1936 historical event was therefore eliminated from inference. The extreme storm
surges occurred during the time-window 1720–1767 will be analyzed and the development of a methodology
to quantify the surges induced by the events from the last part of the 18[th] and the 19[th] century is undergoing.
Table 5 shows quantified water levels (for Dunkirk, Gravelines, Calais, Oostende and Nieuport) compared to
the associated Mean High Water Springs (MHWS) for the 1720–1767 events. The MHWS is the highest level
reached by spring tides (on the average over a period of time often equal to 19 years). De Fourcroy D-R.
(1780) presented the water levels in royal foot of Paris, where 1 foot corresponds to 0.325 m and is divided
into 12 inches (1 inch = 0.027 m) except for the Oostende levels that are given in Flemish Austrian Foot
(corresponds to 0.272 m and is divided in 11 inches).
As a first approach the height of the surge above the MHWS level was estimated, which has the
advantage that the local reference level doesn't need to be transposed into the French leveling system and
as the historic sea level is considered, there is no need to assess sea level rise due du climate change can
be neglected. De Fourcroy D-R. (1780) gave water levels for the five cities in their respective leveling
system: In Calais the zero corresponds to a fixed point on the Citadelle sluice, in Gravelines the zero
corresponds to à fixed point on the sluice of river Aa. For Dunkirk the "likely low tide of mean spring tides" is
considered as a zero point and marked on the docks of Bergues sluice, we will refer to this zero as Bergues
Zero afterwards. The location of the measure point of Bergues Sluice is presented in in Fig. 1 (to the right) on
an old plan of the Dunkirk city. Fig. 6 shows the MHWS water level and the extraordinary water levels for the
storm events of 1720, 1763 and 1767 in Dunkirk.
The difference between the observed water levels and the MHWS is the surge above MHWS. The three
levels are about the same height, ranging from 1,46 m to 1,62 m. We calculated the surge above MHWS for
Calais, Gravelines, Nieuport and Oostende; they're shown in Table 6. It is interesting to note that, for the
1763 and 1767 events, the highest levels were reconstructed in Oostende and the lowest levels in Calais.

### 4.4 Dunkirk surge series

For the sake of convenience and for more precision, we need to refine the surges above MHWS estimated in
the previous section (Fig. 6 and Table 6). This refinement required the development of a tide coefficient
based methodology. Indeed, the tide coefficient for each storm event indicates whether surge above MHWS
is over- or underrated or approximately right. As this coefficient is calculated for the Brest site and applied to
the whole coastal zone, a table showing expected mean levels in Dunkirk for each tide coefficient was
established. One tide coefficient estimated at Brest can have different high water levels at Dunkirk. For this
study, it was assumed that the historic MHWS corresponds to the tide coefficient 95. In the developed
methodology, all the 2016 high tides for each tide coefficient are used and the water levels for each tide
coefficient are averaged. The difference $\Delta_{WL}$ between this averaged level and the water level corresponding
to the tide coefficient 95 (the actual MHWS) is then calculated and added (or subtracted) to the historic surge





above MHWS. In case we have two surges the mean of the two values is considered. Results for the Dunkirk
surges are shown in the last column of table 7.
In addition to the water levels reached during events and in specific years, other types of historical
information (lower bounds and ranges) can be gathered. For instance, De Fourcroy D-R. (1780) stated that
the highest water level measured during the period 1720-1767 was the one induced by the 1767
extraordinary storm. Paradoxical though it may seem at first sight, the skew surge caused by the 1763 storm
is greater than the 1767 one. A plausible explanation is that the 1767 event was occurred when the tide was
higher than that of 1763.
For the Dunkirk series, it is interesting to see that it is easier to quantify events from the 18[th] century, as
the water levels were either measured or reconstructed only a few years after the events took place. During
his thesis, N. Pouvreau (2008) started an inventory of existing tide gauge data available in different archive
services in France. According to him, the first observations of the sea level in Dunkirk were made in the
years 1701 and 1702, where time and height were reported. Observations were also made in 1802 and
another observation campaign was held during 1835. The first longer series is dated from 1865 – 1875. For
the 20[th] century only sparse data is available for the first half of the century. Pouvreau (2008) only listed the
data found in the archives of the National Geographic Institute (Institut Géographique National IGN), the
Marine Hydrographic and Oceanographic Service (Service Hydrographique et Océanographique de la
Marine SHOM) and the Historical Service of Defense (Service Historique de la Défense SHD). During the
present study we found evidence that sea levels were measured at Bergues sluice during the 18[th] century
and that diverse hydrographic campaigns were made during the 19[th] century (De Fourcroy D-R., 1780). This
research and first analysis of historic data shows the potential of the data collected, as we were able to
quantify some historical skew surges, but it also shows how difficult and time consuming the transformation
of descriptive information into skew surge values is and that more detailed analysis will be necessary to
quantify the other historical surge events.

## 5   Impact of the information gathered in the archives on the frequency analysis

The robustness of the POTH FM is one of the more significant issues we must deal with. The main focus of
this discussion is the assessment of the impact of the additional HI (gathered from the archives) on the
frequency estimates for high RLs. The same FM was used but with additional HI and different settings. The
HI gathered from both literature and archives with some model settings are summarized in table 8. The
results of the POTH FM using HI from both literature and archives (called hereafter the full FM) are reported
herein. These results are presented in form of a probability plot (Fig. 7) and a table summarizing the
estimates of the desired RLs and associated CIs (Table 9). Fig. 7 consists of two subplots related to the FA
of the Dunkirk extreme surges. The left side of the figure shows collected data: the systematic surges are
represented with the grey bars, the historical surges extracted from the literature with red bars and those
extracted from the archives (estimated and corrected with regards to the tide coefficients) are represented
with green ones. We can also see the two time-windows (the blue background areas in the graph) 1720-
1770 and 1897-2015 used in the POTH FM as historical periods. The right side shows the results of the full
FM. As mentioned in part 4.3, to consider the full POTH FM, six historical storm surges distributed equally (
$n_k = 3$) over two not-successive time-windows: 1720-1770 ($w_{HMax1} = 50$ years) and 1897-2015 ($w_{HMax2} = 72,5$
years, knowing that $w_s = 46,5$ years) are used as historical data. In the plotting positions, the archival
historical surges are represented by green squares, while those found in the literature are depicted by red
circles. The fitting presented in Fig. 7 shows a good adequacy between the plotting positions and theoretical
distribution function (calculated probabilities of failure). Indeed, all the points of the observed distribution are
not only inside the CI, but even better, they are almost on the theoretical distribution curve. The results of
table 9 show that:
- The RLs of interest had increased by only 10 to 20 cm. This is an important element of robustness. Indeed,
adding or removing one or more extreme values from the dataset does not significantly affect the desired
RLs. In other words, it is important that the developed model is not very sensitive (in terms of RLs used as
design bases) to a modification in the data regarding very few events. As a matter of fact, the model owes
this robustness to the exhaustiveness of the available information.
- The relative widths of CIs with no archival HI included are 1.5 times larger than those given by the full
model. This means that the user of the developed model is more confident in the estimations when using
the additional HI gathered in the archives.
After collecting HI about the most extreme storm surge events in the 18[th] and 20[th] centuries, it was first
found that the 1953 event is still the most important one in term of magnitude. The developed POTH FM
attributed a 200-year return period to this event. The value of the surge induced by the 1953 storm is
between 1,75m and 2,50 m. That said, it is interesting to note that this CI includes the value of 2.40
estimated by Le Gorgeu and Guitonneau (1954). This may be a reason to think that the continuation of our
work on the quantification of the skew surges occurred in the 19[th] century will may be reveal extreme surges
similar to that induced by the 1953 storm.



## 6 Conclusion & perspectives

To improve the estimation of the risk associated to exceptional high surges, historical information about storms and marine flooding events for the Nord-Pas-de-Calais was collected by historians for the 1500-1950 period. Qualitative and quantitative information about all the extreme storms hit the region of interest were extracted from a large number of archival sources. In this paper, we presented the case study of Dunkirk in which the exceptional surge induced by the 1953 violent storm appears as an outlier. In a second step, the information gathered (in both literature and archives) was examined. Quality control and cross validation of the collected information indicates that our list of historic storms is complete as regards extreme storms. Only events occurred in the periods 1720-1770 and 1897-2015 were quantified and used in the POTH FM as historical data. To illustrate challenges and opportunities for using this additional data and analyzing extremes over a longer period than previously possible, the results of the FA of extreme surges was presented and analyzed. The assessment of the impact of additional historical information is carried out by comparing theoretical quantiles and associated confidence intervals, with and with no archival historical data, constitutes the main result of this paper.

The conclusions drawn in previous studies were examined in greater depth in the present paper. Indeed, on the basis of the results obtained previously (Hamdi et al, 2015) and in the present paper, the following conclusions are reached:

- The use of additional historical information over longer periods than the gauging one, can significantly improve the probabilistic and statistical treatment of a dataset containing an exceptional observation considered as an outlier (i.e. the 1953 storm surge).
- As the historical information gathered in both literature and archives tend to be extreme, the right tail distribution has been reinforced and the 1953 "exceptional" event don't appear as an outlier any more.
- As this additional information is exhaustive (relatively to the corresponding historical periods), the RLs of interest had increased very slightly and the confidence intervals were reduced significantly.

An in-depth study could help to thoroughly improve the quantification method of the historical surges and apply the developed model on other sites of interest.

## Acknowledgements
The Authors thank the municipal archives of Dunkerque and Gravelines for the support during the historical information collect.

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





**Table 1** Date, localization, water and surge levels (m) of gathered storms within Nord-Pas-de-Calais area.

| Date | Location | Predicted WL | Observed WL | Surge | Source |
|---|---|---|---|---|---|
| 02/02/1791 | Dunkirk | - - - | - - - | - - - | Newspapers[5] |
| 14/01/1808 | Dunkirk | - - - | WL ~ 02/02/1791 | - - - | - - - |
|  | Walchern Island, NL | - - - | W rose up to 25ft[4] | - - - | Newspapers[5] |
| 19/02/1882 | Sangatte, Calais | - - - | - - - | 1,25[1] | Deboudt, 1997 |
| 28/11/1897 | Malo-les Bains Dunkirk | 5,50[1] | 7,36[1] | 1,86[1] | DREAL Nord – Pas de Calais |
| 07/01/1905 | Sangatte, Calais | 6,80[1] | 7,70[1] | 0,90[1] | Deboudt, 1997 |
| 31/12/1921 | Sangatte, Calais | - - - | - - - | - - - | Deboudt, 1997 |
| 01/03/1949 | Dunkirk | 5,70 NGF | 7,30 NGF[2] | 1,60 | Le Gorgeu & Guitonnau, 1954 |
|  |  |  | 7,55 NGF[2] | 1,85 | DREAL Nord–Pas de Calais |
|  | Antwerpen (BE) | - - - | > 7 TAW[3] | - - - | Codde and De Keyser 1967 |
| 01/02/1953 | Sangatte, Calais | 6,70 | 8,20 | 1,50 | Deboudt, 1997 |
|  | Dunkirk | 5,50 | 7,90 | 2,40 | Le Gorgeu & Guitonnau, 1954 |
|  | Dunkirk | 5,77 | 7,90 | 2,13 | Bardet et al., 2011 |

[1] no reference leveling given; [2] NGF : the French Ordnance Datum (Nivellement Général Français); [3] TAW = Nivellement Belge; [4] no
indication which feet (royal french feet / flemish austrian feet); [5] Newspapers: Journal Politique de Mannheim 26, 30 Janvier 1808 ;

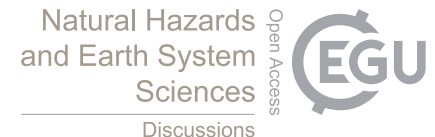

**Table 2** HI extracted from the literature

| Year | 1897 | 1949 | 1953 | 1995 |
|------|------|------|------|------|
| Surge (m) | 1,86 | 1,60 | 2,13 | 1,15 |







**Table 3** The T-year quantiles & relative widths of their 70% CI (all the duration are given in years)

| T (years) | + 1953 (as hist. Data) | | + HI from literature | |
|---|---|---|---|---|
| | $w_{HMax}=16,5$ | | $w_{HMax}=72,5$ | |
| | $S_T$ | $\Delta CI/S_T$ | $S_T$ | $\Delta CI/S_T$ |
| 100 | 1,76 | 40% | 1,82 | 32% |
| 500 | 2,46 | 71% | 2,59 | 56% |
| 1000 | 2,86 | 86% | 3,03 | 69% |




**Table 4** Details of 1500–2015 Nord-Pas-de-Calais historical storms and storm surges sources.

| Year/Date | Data Type | Quality Index | Source Name | Observer occupation |
|---|---|---|---|---|
| 1507 | Surge | 3 | L'Abbé Harrau (1901) | Historian |
| 01/11/1570 | Surge | 3 | Pierre Faulconnier (1730) | Mayor of Dunkirk |
| 1605 | Surge | 3 | Victor Derode (1852) | Historian |
| 12/01/1613 | Surge | 4 | MAS-O (XVIIIth century) - Jean Hendricq bourgeois | Bourgeois and merchant of the city |
| 01/11/1621 | Surge | 4 | Céléstin Landrin (1888) | Archivist (Calais) |
| 03/11/1641 | Surge | 4 | M. Lefebvre (1766) | Priest |
| 1644 | Surge | 3 | Victor Derode (1852) | Historian |
| 1663 | Surge | 3 | Baron C. de Warenghien (1924) | Historian |
| 12/1663 | Surge | 3 | | |
| 1665 | Surge | 3 | Victor Derode (1852) | Historian |
| 1671 | Surge | 3 | | |
| 1675 | Surge | 3 | | |
| 16/02/1699 | Surge | 3 | L'abbé Harrau (1903) | Historian |
| 1715 | Surge | 3 | Victor Derode (1852) | Historian |
| 1720 | Surge | 4 | De La Lande (1781) | Astronomer |
| 31/12/1720 | Surge | 3 | Charles Demotier (1856) | Local Historian |
| 25/12/1738 | Storm | | | |
| 1734 | Surge | 4 | MAD (AncDK15) | Unknown |
| 19/01/1735 | Storm | | | |
| 27/02/1736 | Surge | 4 | MAD, (AncDK291)/C. Demotier (1856) | Historian |
| 01/10/1744 | Storm | 3 | Jean Louis le Tellier (1927) | Local of Dunkirk |
| 11/03/1750 | Surge | 3 | De La Lande (1781) | Astronomer |
| 06/07/1760 | Storm | 3 | Almanach de Calais (1845) | Unknown |
| 02/12/1763 | Surge | 3 | De La Lande (1781) | Astronomer |
| 28/09/1764 | Surge | 2 | J. Goutier «Amis du Vieux Calais» | Unknown |
| 02/01/1767 | Surge | 4 | M.A. Bossaut (1898) | Librarian |
| 05/1774 | Surge | 3 | MAD, ref. 2 Fi 169 | Unknown |
| 01/01/1777 | Surge | 3 | Raymond de Bertrand (1855) | Writer |
| 01/01/1778 | Storm | 3 | Leon Moreel (1931) | Lawyer |
| 31/12/1778 | Surge | 4 | Pigault de Lespinoy, 19th cent. - a | Mayor of Calais |
| 02/02/1791 | Surge | 4 | Pigault de Lespinoy, 19th cent. - b | |
| 17/11/1791 | Surge | 2 | Bernard Barron (2007) | Journalist |
| 04/09/1793 | Surge | 3 | L'abbé Harrau (1898) | Historian |
| 30/10/1795 | Storm | 3 | Céléstin Landrin (1888) | Archivist (Calais) |
| 13/11/1795 | Storm | 3 | Charles Demotier (1856) | Historian |
| 09/11/1800 | Surge | 4 | MAD, ref. 2Q9 | Unknown |
| 29/03/1802 | Storm | 3 | Augustin Lemaire (1857) | Regent |
| 03/11/1804 | Storm | 3 | Augustin Lemaire (1857) | Regent |
| 1807 | Surge | 3 | Victor Derode (1852) | Historian |
| 18/02/1807 | Storm | 3 | Mannheim, 26/01/1808 | Newspaper |
| 02/12/1807 | Storm | 3 | Augstin Lemaire (1857) | Regent |
| 14/01/1808 | Surge | 4 | MAC, «floods» sheets | Archivists (Dunkirk) |
| 14/11/1810 | Storm | 3 | Christian Gonsseaume (1988) | Historian |
| 03/01/1825 | Surge | 2 | MAC, «storms» sheets | Archivists (Dunkirk) |
| 04/02/1825 | Surge | 4 | MAD, ref. 5O6 | Harbor Engineer |
| 19/10/1825 | Storm | 4 | MAC, «storms» sheets | Archivists (Dunkirk) |
| 29/11/1836 | Surge | 3 | Union Faulconnier(1936) | Mayor of Dunkirk |
| 02/01/1846 | Surge | 3 | Victor Derode (1852) | Historian |
| 02/10/1846 | Surge | 3 | | |
| 26/09/1853 | Storm | 3 | | |
| 26/10/1859 | Storm | 3 | Dr. Zandyck (1861) | Military Surgeon & Physician |
| 02/11/1859 | Storm | 3 | | |
| 16/01/1867 | Storm | 2 | Gilles Peltier «Amis du Vieux Calais» | Unknown |
| 02/12/1867 | Storm | 2 | Bernard Barron (2007) | Journalist |
| 30/01/1877 | Storm | 4 | MAC, «storms» sheets | Archivists (Dunkirk) |
| 21/12/1892 | Storm | 3 | Céléstin Landrin (1888) | Archivist (Calais) |
| 10/01/1893 | Storm | 3 | MAD, reference 5 S 1 | Harbor Engineer |
| 18/11/1893 | Storm | 2 | Gilles Peltier «Amis du Vieux Calais» | Unknown |
| 11/10/1896 | Storm | 2 | Christian Gonsseaume (1988) | Historian |
| 27/01/1897 | Storm | 2 | Christian Gonsseaume (1988) | Historian |
| 29/11/1897 | Storm | 2 | MAD, reference 4 S 874 | Architect Gontier |
| 02/03/1898 | Storm | 4 | Le Gravelinois, (19/03/1989) | Unknown |
| 13/01/1899 | Storm | 4 | Le Nord Maritime, (January, 1899) | Unknown |
| 10/12/1902 | Storm | 2 | Christian Gonsseaume (1988) | Historian |
| 11/09/1904 | Storm | 3 | Emile Bouchet (1911) | Man of Letters |
| 08/01/1928 | Storm | 2 | Christian Gonsseaume (1988) | Historian |
| 07/12/1929 | Storm | 2 | Christian Gonsseaume (1988) | Historian |
| 28/11/1932 | Storm | 4 | MAD, ref. 4 S 881 | City council of Dunkirk |
| 01/12/1936 | Surge | 4 | | |
| 01/03/1949 | Surge | 4 | La Voix du Nord, 2-4/03/1949 | Unknown |
| 01/02/1953 | Surge | 4 | La Voix du Nord, 4-6/02/1953 | Unknown |
| 16/09/1966 | Surge | 4 | La Voix du Nord, 17/09/1966 | Unknown |
| 02/01/1995 | Surge | 4 | Maspataud A., (2011) | PhD student |

MAS-O : Saint-Omer Municipal Archives - Historical collection of Jean Hendricq bourgeois of Saint Omer; MAD : Municipal Archives Dunkirk; MAC : Municipal Archives Calais – thematic sheets






**Table 5** Historical information about water levels in Dunkirk and other cities (unless otherwise stated, Heights
are given in French royal foot which corresponds to 0,325 m).

| Date & N° | Tide Coefficient[1] | The event characteristic | Wind direction | City | Water level | Source name |
|---|---|---|---|---|---|---|
| 31/12/1720 | | | | | | De Fourcroy D-R. (1780); |
| 1 | 104-104 | Violent storm | NW | Dunkirk | 22 ft 3 in[**] | Plocq (1873). |
| 27/02/1736 | | | | | | De La Lande, (1781) ; |
| 2 | 110-114 | Generally accompanied by strong winds | SW to N | Gravelines | 13 ft 2 in[**] | De Fourcroy D-R. (1780). |
| | | | | Calais | > 1767 | |
| 11/03/1750 | | | | | | De La Lande, (1781) ; |
| 3 | 115-111 | Generally accompanied by strong winds | SW to N | Gravelines | 12 ft 2 in | De Fourcroy D-R. (1780); |
| | | | | Oostende | 13 ft 6 in[*] | Mann, D. (1777, 1780). |
| 02/12/1763 | | | | | | De La Lande, (1781) ; |
| 4 | 78-81 | Generally accompanied by strong winds | SW to N | Dunkirk | 22 ft | De Fourcroy D-R. (1780); |
| | | | | Calais | 17 ft 2 in | Mann, D. (1777, 1780) |
| | | | | Gravelines | 14 ft 2 in | |
| | | | | Oostende | 14 ft[*] | |
| | | | | Nieuport | 14 ft[*] | |
| 02/01/1767 | | | | | | Histoire de l'Académie Royale des Sciences (1767) ; |
| 5 | 93-96 | Horrible storm | WNW-NNW | Dunkirk | 22 ft 6 in | |
| | | | | Calais | 18 ft 8 in | De Fourcroy D-R. (1780); |
| | | | | Gravelines | 15 ft 10 in | Mann, D. (1777, 1780) |
| | | | | Oostende | 16 ft[*] | |
| | | | | Nieuport | 17 ft 1 in[*] | |
| 01/12/1936 | | | | | | MAD 4S 881 |
| 6 | 99-96 | Violent storm | | Dunkirk | 1 m>pred | |

[*] Source: SHOM; [**] reconstructed water levels; [*] foot of Brussels (1 ft = 0.273 m).



**Table 6** Surges above MHWS (given in meters)

| Date | Calais | Gravelines | Nieuport | Oostende |
|------|--------|-----------|----------|----------|
| 1736 | 1,06[*] | 1,38 | --- | --- |
| 1750 | --- | 1,05 | --- | 1,05 |
| 1763 | 0,57 | 0,97 | 0,97 | 1,10 |
| 1767 | 1,06 | 1,51 | 1,60 | 1,94 |





**Table 7** Historical skew surges induced by the 1720, Heights are given in m

| Date | Tide Coeff. | Surge above MHWS | $\Delta_{WL}$ | Skew surge |
|------|-------------|------------------|---------------|------------|
| 1720 | 104 | 1,54 | -0,17 | 1,37 |
| 1763 | 78/81 | 1,46 | 0,29/0,24 | 1,75/1,7 |
| 1767 | 93 | 1,62 | 0,01 | 1,63 |






**Table 8** The HI dataset (from literature and archives). Surges are given in m and $w_{HMax}$ and $w_s$ in years.

| Year | 1720 | 1763 | 1767 | Events exist ($n_k \neq 0$) but cannot be quantified | 1897 | 1949 | 1953 |
|---|---|---|---|---|---|---|---|
| Surge (m) | 1,37 | 1,75 | 1,63 | | 1,86 | 1,60 | 2,13 |
| | • HI from archives, $n_k = 3$ <br> • 1720-1770 time-window <br> • $w_{HMax1} = 50$ | | | • HI from archives, $n_k \neq 0$ <br> • 1770-1897 time-window <br> • Not used in the inference | • HI from literature, $n_k = 3$ <br> • 1897-2015 time-window <br> • $w_{HMax2} = 72,5$ ; $w_s = 46,5$ | | |





**Table 9** The T-year quantiles & relative widths of their 70% CI (all the duration are given in years)

| T (years) | +1953 event | | + literature HI | | + literature & archives HI | |
|---|---|---|---|---|---|---|
| | $w_{HMax1} = 16,5$ | | $w_{HMax} = 72,5$ | | $w_{HMax1} = 50$ ; $w_{HMax2} = 72,5$ | |
| | $S_T$ | $\Delta CI/S_T$ | $S_T$ | $\Delta CI/S_T$ | $S_T$ | $\Delta CI/S_T$ |
| 100 | 1,76 | 40% | 1,82 | 32% | 1,84 | 26% |
| 500 | 2,46 | 71% | 2,59 | 56% | 2,61 | 48% |
| 1000 | 2,86 | 86% | 3,03 | 69% | 3,05 | 59% |





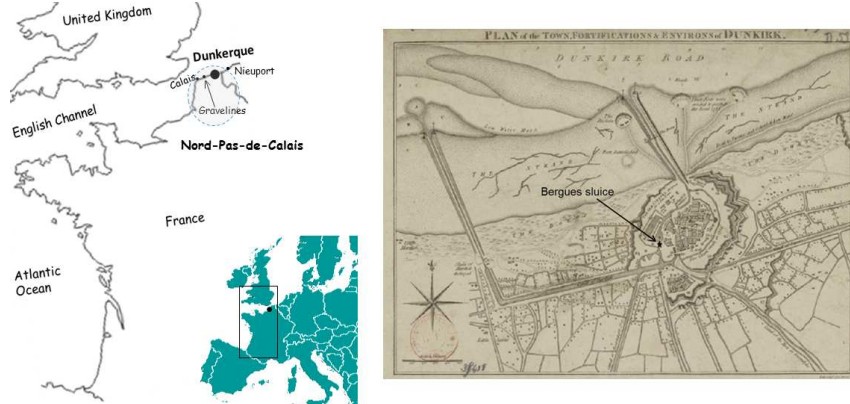

**Fig. 1.** Map of the location (to the left) and an old plan of the Dunkirk city with the measure point of Bergues Sluice (to
the right)




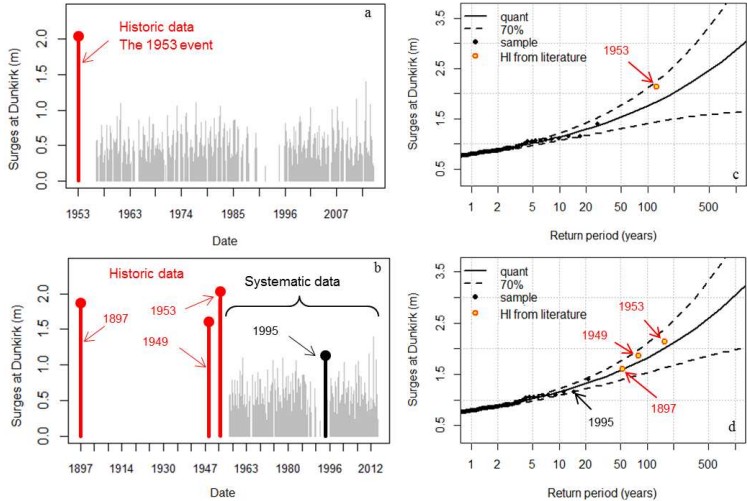

**Fig. 2.** The GPD fitted to the POTH surges in Dunkirk: with the 1953 event as a historical data (top panel); with
historical data from literature (bottom panel). The 1995 event is considered as systematic.




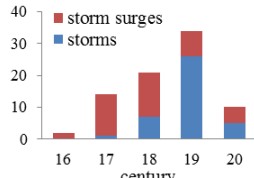 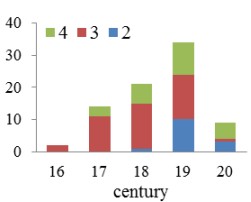


**Fig. 3.** Distribution in time and type of the events in the data base (left); Quality of the data. For each event the best source has been classified (right).





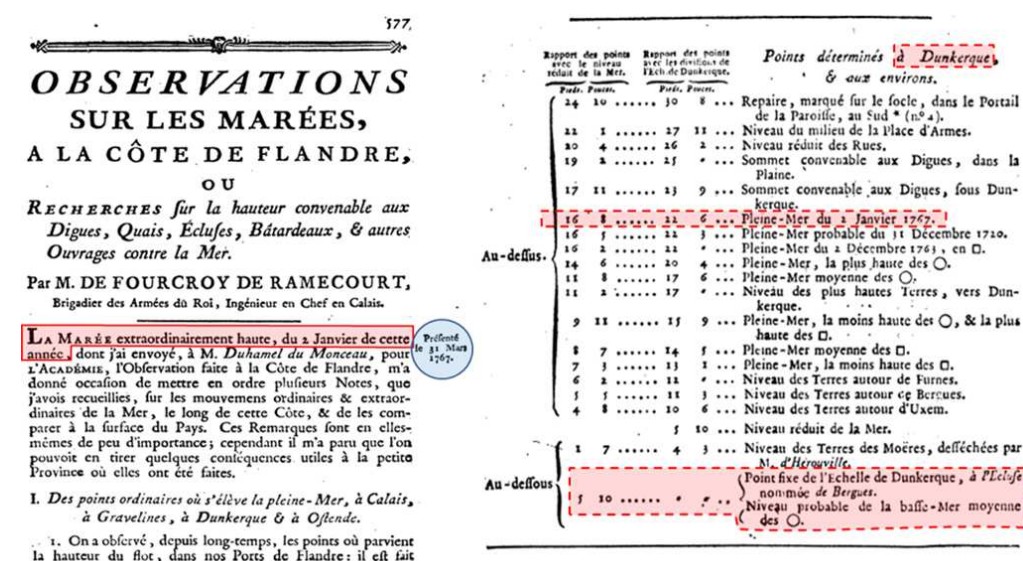

**Fig. 4.** HI (as presented in the archives) about the 1767 extreme surge event in Dunkirk (De Fourcroy D-R., 1780)




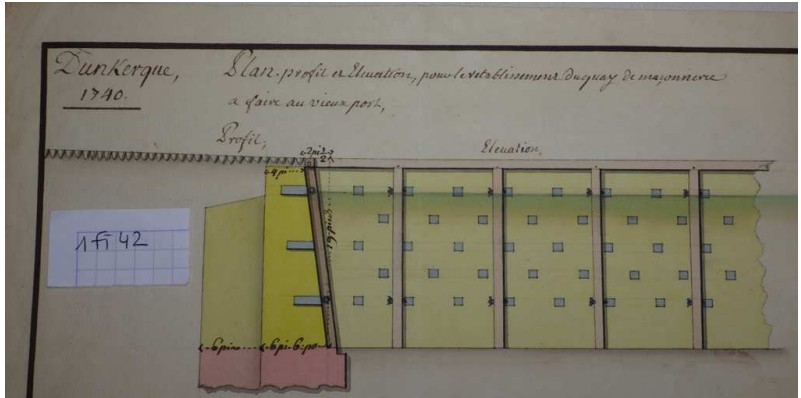


**Fig. 5.** A profile of the Dunkirk harbor dock (the municipal archives of Dunkirk – ref. 1Fi42, 1740).





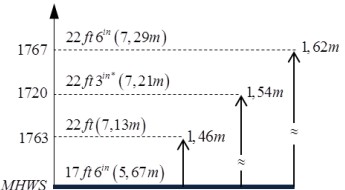


**Fig. 6.** Water levels in relation to the measure point of Bergues Sluice in Dunkirk and surges above MHWS.





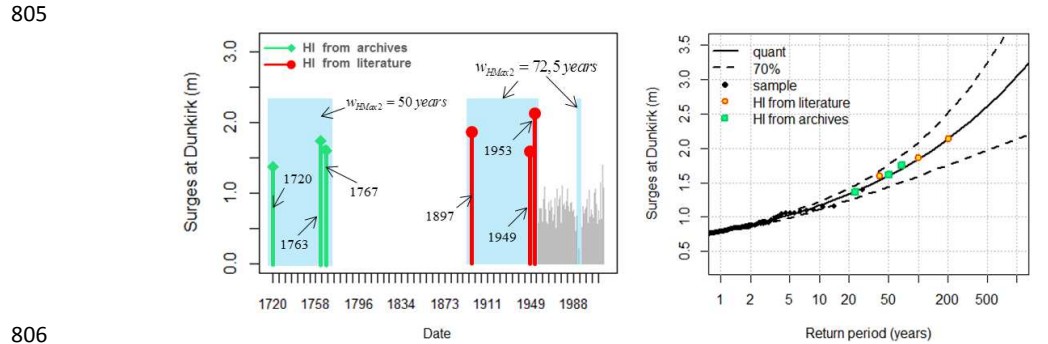

**Fig. 7.** Historical and Systematic Skew surges in Dunkirk and model settings (left); The GPD fitted to the POTH surges
in Dunkirk (right)