# Peer review of "Analysis of the risk associated to coastal flooding hazards: A new historical extreme storm surges dataset for Dunkirk, France"

_Natural Hazards and Earth System Sciences, 2017_

## Referee Comment (RC1) · Anonymous Referee #1 · 5 Jan 2018

***** General comments

It is well known that deriving water levels for large return periods using limited duration of observation can result in very significant errors. In this paper, the authors attempted to address this issue for the city of Dunkirk through the collection of long-term historical data. Although this approach is not really a first of its kind, it is still extremely interesting and valuable, and as such, it could be considered favorably for publication, provided that the authors address the two main concerns (and minor issues, see below) that I have regarding this paper:

1-The authors state in the abstract that the aim is to demonstrate the technical feasibility of including long-term historical information to improve the statistical assessment of extreme water level return periods. But what is reallyÂătechnically challenging when doing this? It is indeed challenging to find old HI, and the authors really deserve to be thanked for this valuable effort. But what is at least as challenging (in my opinion) is to answer this kind of questions:

-to what extent are the storm surge dynamics that occurred hundreds of years ago representative of the actual level of risk?

-Has the bathymetry, topography, or land cover of the studied area evolved since then, and what could be the impact on storm surge dynamics?

-How accurate are the historical water level data, considering, for example, potential sea level rise, land subsidence, uncertainties relative to the distinction between over-flowing and overtopping when assessing maximum water levels, etc?

Answering these questions could be extremely difficult, but I think that the authors should at least discuss them in the paper.

2-This study (as others) relies on one extremely strong hypothesis: the maximum wa-ter level is supposed to be spatially homogeneous, not only in Dunkirk, but also for the nuclear plant in Gravelines, 20km away. To what extent can this assumption be con-sidered realistic? For example, tide gauges generally do not capture the whole wave setup component of the surge, which can vary by a few dozens of centimeters between a harbor, and nearby beaches. They also generally poorly capture infragravity waves, which have been observed in many places along the shoreline of France and can have huge impacts on coastal flooding. The bathymetry and topography can be also quite different in Dunkirk, and 20km away from the city. Considering the very high stakes, it is hence important I think to address these issues (or at least to discuss them) in the paper.

****** Specific comments

Line 23: "Dunkirk site, representative of the Gravelines NPP": this statement should be tempered or discussed in greater details, as mentioned above.

Lines 117-119: it is unclear whether or not you have already built a complete historical database for the entire French coasts (at least Atlantic and English Channel). Please reformulate.

Table 1: please specify (when possible) where exactly the water and surge levels were obtainedĂă(at tide gauges? Dikes? in the streets? In houses? in areas exposed to waves or not?). It would be also interesting to know for each case to what extent the values could be affected or not by wave setup, wave run-up, or overtopping. If an area at a given altitude is flooded because of overtopping, it should not be treated the same way as if it was flooded by overflowing for example).

Please do the same whenever possible for Tables 4-5

Lines 267-268: "A POT threshold equal to 0,75m [. . .] is an adequate choice". Please give at least a few indications or a reference to explain how you came to choose this value.

Lines 445-446: this seems to suggest that some historical data (for the 18th century for example) have been collected but were not used in this paper. Why? If the purpose of this study is really to demonstrate the technical feasibility of a long-term historical study, then it is more important to describe this kind of information than computing new extreme water levels. On the other hand, if the objective is to do these computations, then all the available data have to be taken into account.

****** Typographical-technical corrections

There are a number of typographical corrections that need to be done. You will find a few examples below. Please carefully proofread the paper again before submitting a revised version.

The term "marine flooding" or "marine submersion" can be found here in many sentences. Although it is used in a number of papers (especially in those written by french-speaking authors), it is not always considered as "proper english". It might be preferable to use "coastal flooding" or "storm surge" instead, as in the title.

Line 17: please define HI in the abstract

Line 39: does not

Line 59: if they had occured

Line 72: why "unfortunately"? Outliers are information. As you mention in the paper, having an outlier indicates that a simple extreme analysis using limited water level data might induce large errors. Better to know that beforehand.

Line 116: have encouraged

Line 117: that occured

Line 120: is to collect HI about storms and storm surges that occurred

Line 159: that occurred

Line 166 and elsewhere: please choose between "Dunkerque" and "Dunkirk" in the paper, or indicate clearly in the introduction that both names refer to the same location.

Lines 195-196: exceeded 2.25m, reaching 3.90m at Harlingen, Netherlands. Large areas...

Line 202, 371, and others: therefore

Line 323: have a lower

Lines 325-326: used in the (. . .?) of

Line 361: that occurred

Lines 668-673: the reference "Hamdi et al 2014" is given twice

[Figure]

Line 740: please indicate in english the meaning of TAW

---

## Referee Comment (RC2) · Anonymous Referee #2 · 26 Jan 2018

General Comments

The manuscript nhess-2017-417 presents the reconstruction of the storm surge level in Dunkirk, utilizing data from different sources and dating back to the 16th century. It is a remarkable effort towards reconstructing the storm surge climate in Dunkirk and the detailed literature review provided is of invaluable significance.

The current form of the manuscript requires major revision since the syntax of the language is often problematic. The incoherent structure throughout the text and especially the description of the results, along with the poor quality of the presented results, makes it difficult for the reader to follow. The publication has the potential to be use-

ful for future studies related with the impact of coastal floods, as soon as a proper justification of some technical approaches is provided.

Specific Comments

As stated above, the description of the data needs to be improved – see also some recommendations at the next section. In section 4.3 the historical surge dataset is presented, but it is not clear whether the hydrodynamic component under study is the storm surge level or the total water level (including the contribution of other hydrodynamic components). Additionally, it is not clear how the storm surge level is estimated when only the meteorological conditions are available from the historical records.

While it is a fair assumption that during a storm event, the water level along neighbouring areas may exhibit a similar level, the local bathymetric features and the man-made structures may alter the local water level. Therefore these data should be considered only as qualitatively accurate and not quantitatively. Should these data be used, a comparison with numerical simulations would decrease the level of uncertainty.

Technical Corrections

The MS should be proofread by a native English speaker for errors in syntax, grammar, spelling and vocabulary.

Informal expressions and language (e.g. "an important surge", "horrible storm") are used for the context of a scientific journal, while the terminology is not the most appropriate (e.g. "marine flooding", "marine submersion").

The manuscript lacks structure and is very difficult for the reader to follow, as the presentation of the data takes place together with the analysis. It is recommended first to describe the data that will be analyzed; this section should be followed by a short description of the methods and finally a section that presents the results after incorporating all the three types of available data.

Although the POTH method has been described in previous publications, it is recom-

mended to provide a short summary at the Methods section. This would give a better overview to the reader, regarding the analysis of the data and would enhance the clarity of the paper.

There is an abundance of information (the damage and the fatalities triggered by the storm, the weather description, etc) scattered around the essay that is loosely connected to the main argument. It would be helpful to move this to a supplementary material section; this would tidy up the main points and would make the argument read in a clearer way. For the same reason, measurements obtained from other sites may be omitted too from the main body of the manuscript, since they are not considered at the analysis (e.g. the section from line 206 to 213).

Please provide a map displaying all the places mentioned at the MS.

Section 2 lacks structure, coherence and paragraph unity. The main title of the section as well as the ones of the following paragraphs are misleading and do not correspond to the topic of the paragraphs. Additionally, section 2.2.1 should be renumbered to 2.1.1 as it refers to the tide gauge record and not to the short-term HI.

The quality of the figures, the tables and their captions is poor and should be improved. Fig.6 does not provide any extra information to the reader.

Consider merging Tables 1, 2, 4 preferably presenting only the information related with the storm surge level and the data included in the analysis. All the information with respect to the sources, meteorological conditions etc should be provided in a tabular form at the supplementary material for future reference and for reproducing the analysis of this study.

Minor corrections

Use one type of decimal notation. Sometimes Dunkerque is used instead of Dunkirk. Many missing punctuation marks along the MS (e.g. line 208, 264 and elsewhere).

Line by line review

Line 17: Abbreviation for HI

Line 21: ... was recovered

Line 74: add "sustain" after "is designed to"

Line 114 and elsewhere: better to use "collect" instead of "gather"

Line 131: provide references for the source of uncertainty

Lines 135-136: probabilistic engineers does not seem correct

Line 140: from tide gauge

Line 146: measurements

Line 150: replace "respecting" with "with respect to"

Line 158 to 160: consider revising the sentences "They may be ... themselves non-independent events"

Line 168: as an extreme event

Line 187: a synoptic reconstruction

Line 200: what is the developed regional model, please provide more information

Line 202 and elsewhere: therefore

Line 210: Provide references or refer to the table

Line 228: the maximum water level

Line 236: are not informed – consider revising

Line 242: phrase "while trying to inventory" is not syntactically correct

Line 245: phrase "we are fairly trustful" is not valid

Lines 252 and 253: consider revising the phrase "Ransom that choice ... in the past"

Line 285: please provide reference for the 1995 skew surge

Line 293: Generalized Pareto Distribution

Lines 318 to 322: consider revising the phrase "This assessment to 20-25% narrower"

Line 391: nature instead of natures

Line 397: replace "memories" with e.g. "eyewitness statements"

Line 422: remove "below"

Line 435: it is the water level during spring tide that was increased and not the spring tide

Line 477: is it "at last" or "at least"

Line 486: what do you mean by "quantified water levels" ?

Line 494: remove "du"

Line 497: to "a" fixed point

Line 545: capitalize "table"

Line 598: "does not" instead of "don't"

Line 739 and Table 1: Walcheren island

---

## Author Comment (AC1) · 16 Feb 2018

February 16, 2018

Re: Resubmission of manuscript "Analysis of the risk associated to coastal flooding hazards: A new historical extreme storm surges dataset for Dunkirk, France", nhess-2017-417

Copernicus Publications
Editorial Support

Dear Editor,

Thank you for the opportunity to revise our manuscript, "Analysis of the risk associated to coastal flooding hazards: A new historical extreme storm surges dataset for Dunkirk, France". We appreciate the careful review and constructive suggestions. It is our belief that the manuscript is substantially improved after making the suggested edits.

Following this letter are the reviewers comments with our responses, including how and where the text was modified. As suggested by the reviewers, our manuscript had been checked by a professional English editing service. The revision has been developed in consultation with all coauthors, and each author has given approval to the final form of this revision.

Thank you for your consideration.

Sincerely,

Yasser Hamdi

**Point-by-Point response / reviewer # 1**
Yasser Hamdi

**General comments**

| Comments | Responses to comments |
|---|---|
| It is well known that deriving water levels for large return periods using limited duration of observation can result in very significant errors. In this paper, the authors attempted to address this issue for the city of Dunkirk through the collection of long-term historical data. Although this approach is not really a first of its kind, it is still extremely interesting and valuable, and as such, it could be considered favorably for publication, provided that the authors address the two main concerns (and minor issues, see below) that I have regarding this paper:
 1. The authors state in the abstract that the aim is to demonstrate the technical feasibility of including long-term historical information to improve the statistical assessment of extreme water level return periods. But what is really a technically challenging when doing this? It is indeed challenging to find old HI, and the authors really deserve to be thanked for this valuable effort. But what is at least as challenging (in my opinion) is to answer this kind of questions:
 - to what extent are the storm surge dynamics that occurred hundreds of years ago representative of the actual level of risk?
 - has the bathymetry, topography, or land cover of the studied area evolved since then, and what could be the impact on storm surge dynamics?
 - how accurate are the historical water level data, considering, for example, potential sea level rise, land subsidence, uncertainties relative to the distinction between overflowing and overtopping when assessing maximum water levels, etc?
 Answering these questions could be extremely difficult, but I think that the authors should at least discuss them in the paper. | What is really a technically challenging is :

 - Find the right source of information, cross with other sources (to find the same information elsewhere and if the event is described in the same way or not). It is then necessary to quantify the information (estimate the value of the storm surge from qualitative information and quantitative information about other physical quantities that can lead to the estimation of storm surges

 - There are several types of historical data (range, exact value, lower bound, threshold of perception). Transform the historical information to these different types is not always easy…

 - In the statistics, one must calculate the empirical probabilities of the historical data (which is not the same as the systematic ones), calculate a likelihood taking into account the heterogeneous data (the historical data) and assign an effective duration to the collected data which an important setting in the frequency model that will be used to estimate the high return levels ...

 - Ensuring the completeness of the information is a task that requires a remarkable effort especially by the historian |
| 2-This study (as others) relies on one extremely strong hypothesis: the maximum water level is supposed to be spatially homogeneous, not only in Dunkirk, but also for the nuclear plant in Gravelines, 20km away. To what extent can this assumption be considered realistic? For example, tide gauges generally do not capture the whole wave setup component of the surge, which can vary by a few dozens of centimeters between a harbor, and nearby beaches. They also generally poorly capture infragravity waves, which have been observed in many places along the shoreline of France and can have huge impacts on coastal flooding. The bathymetry and topography can be also quite different in Dunkirk, and 20km away from the city. Considering the very high stakes, it is hence important I think to address these issues (or at least to discuss them) in the paper. | Indeed, the assumption is strong and the state of current practice is to neglect some local effects… Nevertheless, the spatial homogeneity assumption was not used arbitrarily in the present study (and in other studies dealing with the use of information in a regional context). Indeed, It was concluded in two regional frequency models (which were developed to estimate extreme storm surges in Dunkirk and other sites) that the Gravelines NPP is located in a physically (based the calculation of the extremal dependence coefficient) and statistically homogeneous region centered on the Dunkirk harbor. |

**Specific Comments**

| Section | Comment | Response to reviewer | Response in the paper |
|---|---|---|---|
| Abstract & Introduction (§2) | Line 23: "Dunkirk site, representative of the Gravelines NPP": this statement should be tempered or discussed in greater details, as mentioned above. | All the studies about the Gravelines NPP use data from the Dunkirk harbor. Indeed, in the nuclear safety field, the representativeness of stations (for rainfall, discharges, sea levels, etc.) is being taken quite seriously. An in-depth study comparing sea levels (and storm surges) in Dunkerque and Gravelines (a short series) has been shown that the impact of the local effects is not significant. | A sentence (justifying the use of the Dunkirk harbor to analyze storm surges in Gravelines) was added in the 2nd § of the introduction. |
| 1. the § before the last one | Lines 117-119: it is unclear whether or not you have already built a complete historical database for the entire French coasts (at least Atlantic and English Channel). Please reformulate. | This database is completed and is currently the subject of a working group involving several French organizations to share, complete and maintain it. | The sentence was reformulated. |
| 2.2.2 Table 1

4.2 & 4.3 Tables 4-5 | Table 1: please specify (when possible) where exactly the water and surge levels were obtained (at tide gauges? Dikes? in the streets? In houses? in areas exposed to waves or not?). It would be also interesting to know for each case to what extent the values could be affected or not by wave setup, wave run-up, or overtopping. If an area at a given altitude is flooded because of overtopping, it should not be treated the same way as if it was flooded by overflowing for example). Please do the same whenever possible for Tables 4-5. | Unfortunately, we do not have these details in the archives. | |
| 3.1 | Lines 267-268: "A POT threshold equal to 0,75m […] is an adequate choice". Please give at least a few indications or a reference to explain how you came to choose this value. | OK | The 1st § in section 3.1 was modified. |
| 4.3 | Lines 445-446: this seems to suggest that some historical data (for the 18th century for example) have been collected but were not used in this paper. Why? If the purpose of this study is really to demonstrate the technical feasibility of a long-term historical study, then it is more important to describe this kind of information than computing new extreme water levels. On the other hand, if the objective is to do these computations, then all the available data have to be taken into account. | It is currently the subject of a second article that will be written by the historian and colleagues. | |

**2. Typographical-technical corrections**

All minor corrections proposed by the reviewer were accepted and performed directly in the paper. The line by line review was considered as well.

**Point-by-Point response / reviewer # 2**
Yasser Hamdi

**General comments**

| Comments | Responses to comments |
|---|---|
| The manuscript nhess-2017-417 presents the reconstruction of the storm surge level in Dunkirk, utilizing data from different sources and dating back to the 16th century. It is a remarkable effort towards reconstructing the storm surge climate in Dunkirk and the detailed literature review provided is of invaluable significance. | Indeed, the effort to build the database is important (thanks to the historian with whom we worked). |
| The current form of the manuscript requires major revision since the syntax of the language is often problematic. The incoherent structure throughout the text and especially the description of the results, along with the poor quality of the presented results, makes it difficult for the reader to follow. The publication has the potential to be useful for future studies related with the impact of coastal floods, as soon as a proper justification of some technical approaches is provided. | - A major revision of English for errors (syntax, grammar, spelling and vocabulary) was made;

- The structure and the quality of the presented results have been improved. |

**Specific Comments**

| Comment | Response to reviewer | Response in the paper |
|---|---|---|
| The description of the data needs to be improved – see also some recommendations at the next section. In section 4.3 the historical surge dataset is presented, but it is not clear whether the hydrodynamic component under study is the storm surge level or the total water level (including the contribution of other hydrodynamic components). | As it can be seen in the list of historical data obtained (Tables 1, 2, 6, 7), we seek to estimate the storm surges in the end. Total seal levels are also collected when available. Storm surges are then deducted from total levels by subtracting the predicted levels. | Sentence added to text in the 1st § of section 3.2. |
| Additionally, it is not clear how the surge level is estimated when only the meteorological conditions are available from the historical records. | Only when the tidal coefficient is given, using the approach described in §2 - section 3.3, one can estimate a value of the surge. | |
| While it is a fair assumption that during a storm event, the water level along neighbouring areas may exhibit a similar level, the local bathymetric features and the man-made structures may alter the local water level. Therefore these data should be considered only as qualitatively accurate and not quantitatively. Should these data be used, a comparison with numerical simulations would decrease the level of uncertainty. | Indeed, these data are considered qualitatively accurate and quantitatively uncertain. Although the distance between the places where information were collected and our point of interest (the target site: the Graveline NPP) is very small, the impact of certain local effects can creep in the inference. A relatively short series on the target site is used to compare the extreme levels at this site and Dunkirk. The comparison shows that in most cases the impact is not significant. Otherwise, the comparison with the simulations will make it possible to appreciate this uncertainty. | |

**Technical correction**

| Comment | Response to reviewer | Response in the paper |
|---|---|---|
| The MS should be proofread by a native English speaker for errors in syntax, grammar, spelling and vocabulary. | Ok. | |
| Informal expressions and language (e.g. "an important surge", "horrible storm") are used for the context of a scientific journal, while the terminology is not the most appropriate (e.g. "marine flooding", "marine submersion"). | "marine flooding" and "marine submersion is replaced by "coastal flooding". | "marine flooding" and "marine submersion is replaced by "coastal flooding". |
| The manuscript lacks structure and is very difficult for the reader to follow, as the presentation of the data takes place together with the analysis. It is recommended first to describe the data that will be analyzed; this section should be followed by a short description of the methods and finally a section that presents the results after incorporating all the types of available data. | Structure reviewed. | The paper is restructured as proposed by the reviewer. |
| Although the POTH method has been described in previous publications, it is recommended to provide a short summary at the Methods section. This would give a better overview to the reader, regarding the analysis of the data and would enhance the clarity of the paper. | A general description of the POTH model and settings was provided in sections 4 and 4.3. Ok for further description. | A section 4.2 "The POTH frequency model" is added. |
| There is an abundance of information (the damage and the fatalities triggered by the storm, the weather description, etc) scattered around the essay that is loosely connected to the main argument. It would be helpful to move this to a supplementary material section; this would tidy up the main points and would make the argument read in a clearer way. For the same reason, measurements obtained from other sites may be omitted too from the main body of the manuscript, since they are not considered at the analysis (e.g. the section from line 206 to 213). | - The §s on the description of HI are moved to an appendix. - Description of the measurements which are not considered in the analysis (like the 1808 event) is removed from the main body of the manuscript to the appendix. | Please see Appendix 1. |
| Please provide a map displaying all the places mentioned at the MS. | OK | Fig.1 updated. |
| Section 2 lacks structure, coherence and paragraph unity. The main title of the section as well as the ones of the following paragraphs are misleading and do not correspond to the topic of the paragraphs. Additionally, section 2.2.1 should be renumbered to 2.1.1 as it refers to the tide gauge record and not to the short-term HI. | Structure reviewed. | The paper is restructured as proposed by the reviewer. |
| The quality of the figures, the tables and their captions is poor and should be improved. Fig.6 does not provide any extra information to the reader. | Quality of figures and tables reviewed. Fig. 6 removed; Fig. 4 & 5 improved and merged; Fig. 2 and 7 improved and merged. | |
| Consider merging Tables 1, 2, 4 preferably presenting only the information related with the storm surge level and the data included in the analysis. All the information with respect to the sources, meteorological conditions etc should be provided in a tabular form at the supplementary material for future reference and for reproducing the analysis of this study. | Table 2 removed (merged with Table 1) Tables 5 and 6 merged | |

**Minor corrections Line by line review**

All minor corrections proposed by the reviewer were accepted and performed directly in the paper. The line by line review was considered as well.

---

## Author Response (AR2)

**Point-by-Point response / reviewer # 1**
Yasser Hamdi

**General comments**

| Comments | Responses to comments |
|---|---|
| The authors corrected a number of syntax errors in the new version, and answered to several questions.

However, my main concerns remain, and are even amplified. To put it in a nutshell, after reading both the manuscript and response to reviewers, I realize that I am not able to answer this simple question: what is really the benefit of this paper for the scientific community?

I think that this work would be helpful (and thus could be published) if at least one of the three following conditions was satisfied: | A 1st paper was published to show the usefulness of historical information (on sea levels, storm surges and coastal flooding) in the frequency estimations of extreme storm surges in the La Rochelle region (Hamdi et al., 2015) and it was concluded that a more exhaustive searching for historical information with the help of historians is necessary. We have therefore started an innovative project to collect the historical information about all the extreme events occurred in the Dunkirk area. A great deal of qualitative and quantitative information about sea levels, storm surges and coastal flooding events in Dunkirk were gathered by historians. Lastly, despite the difficulties of validation and quality control, we obtained the old data presented in the paper and we integrated them in the statistical modeling.

The first results of this work are presented in this article. The very important question of the completeness of the information is almost solved in the present work; the confidence intervals have been reduced significantly. Just look at how the fit has improved from one line to another in Figure 4. What is new (apart from the historical information collect) compared to the previous work is the robustness of the results and the best quality of the fit. We are more confident in the estimations. For us (in the nuclear safety field), this is of great importance because we are improving the estimate of the risk associated to coastal flooding.

*The reviewer: What is really the benefit of this paper for the scientific community?*

The present paper has two key benefits for the scientific community.

1- Engineers who must size coastal works in the Dunkirk area, they now have a much better idea about extreme levels to use. We know now for example that the 1953 exceptional storm surge (considered by many scientists and by the media as the one never seen in the region) occurs once every 200 years (in average), according to our calculations. **The data reconstructed using the historical information as well as the results of the statistical analysis presented in our paper are currently used by scientists working in the nuclear safety field in France**.
2- To show that a great deal of information about very old coastal flooding hazards were found in the archives (and where are they!). And also to be aware of the importance of the new data and its influence on extreme levels assessed with statistical methods. |
| 1-if the method was a first of its kind. Unfortunately, using historical information over long periods and showing that outliers are not exceptional has already be shown previously (e.g. Bulteau et al 2015, Hamdi et al 2015). | *The reviewer: if the method was a first of its kind.*

• The collect of historical information about sea levels, storm surges and coastal flooding events in Dunkirk is a first of its kind and as mentioned above, it is of great importance for scientists working for the safety of the Gravelines nuclear power plant for instance.
• The frequency analysis performed in this paper is a first of its kind because a particular work was performed to have some continuity in the data ensuring a better completeness of the added information. The frequency analysis in this paper is then performed without making an assumption about exhaustiveness. |

2-if the authors were really addressing " the technical feasibility" of using long-term historical information to improve the statistical assessment of extreme water level return periods, as they suggest in the abstract. Unfortunately, as I already mentioned in the first review, nothing is said about some of the main challenges for this kind of approach: how to deal with old data uncertainties? How to deal with the evolution of bathymetry, topography, land cover of the studied area? To what extent can we be sure that events which occurred hundreds of years ago are representative of the actual risk level? No new information/discussion is brought on these topics in this paper.

*The reviewer: if the authors were really addressing " the technical feasibility" of using long-term historical information to improve the statistical assessment of extreme water level return periods, as they suggest in the abstract.*

"technical feasibility" (the term used in the abstract), refers to what was really a technically challenging:

- Find the right archive, cross with other sources (to find the same information elsewhere and if the event is described in the same way or not). It is then necessary to quantify the information (estimate the value of the storm surge from qualitative information and quantitative information about other physical quantities that can lead to the estimation of storm surges

- There are several types of historical data (**range**, exact value, **lower bound**, **threshold of perception**). Transform the historical information to these different types is not always easy…

- Ensuring the completeness of the information is a task that requires a remarkable effort especially by the historian.

*The reviewer: How to deal with old data uncertainties?* A review of the literature on HI and the role it can play in a frequency analysis has been made by several authors (e.g., Stedinger and Baker, 1987 - Salas et al., 1994 - Ouarda et al., 1998).

Old data are often imprecise, and we should consider their inaccuracy in the analysis (by using a threshold of perception, range and lower bound data, etc). However, as it was shown in the literature, even with important uncertainty, the use of old data improves significantly the frequency estimations of extreme and rare events and it is a viable mean to decrease the influence of outliers by increasing their representativeness in the sample (Hosking and Wallis, 1986a - Wang, 1990 - Salas et al., 1994 - Payrastre et al., 2011).

Our objectives have been defined based on this point which seems to be a key element for the understanding of our work. The purpose of the paper was to collect the good information and to quantify it in order to obtain approximate values of the variable of interest (storm surge), without seeking accurate magnitudes. However, there are other inputs that must be used with reasonable accuracy: the date (the year) of occurrence of the events as well as the systematic and historical durations, the POT threshold as well as the threshold of perception, etc.). The main goal of our work is to examine the potential gain in estimation accuracy with the use of old data even if it is uncertain.

This was explained in §5 (section: Introduction):

"Because HI is often imprecise, its inaccuracy should be considered in the analysis. Nevertheless, the influence of an outlier can be decreased by increasing its representativity in the sample when using the HI, knowing that its uncertainty is sometimes considerable (e.g. Payrastre et al, 2011; Hamdi et al, 2015)".

For more utility and clarity, authors propose to replace this § by the following one:

"Data reconstructed using historical information are often imprecise, and we should consider their inaccuracy in the analysis (by using a threshold of perception, range and lower bound data, etc). However, As it was shown in the literature, even with important uncertainty, the use of HI is a viable mean to decrease the influence of outliers by increasing their representativeness in the sample (Hosking and Wallis, 1986a - Stedinger and Baker, 1987 - Wang, 1990 - Salas et al., 1994 - Ouarda et al., 1998 - Payrastre et al., 2011).

The reference (Wang, Q.J., 1990) has been added in the references section.

| | |
|---|---|
| | *The reviewer: To what extent can we be sure that events which occurred hundreds of years ago are representative of the actual risk level? How to deal with the evolution of bathymetry, topography, land cover of the studied area?*

Some tests and analyses were conducted to compare old and new data, old and recent conditions and to identify what could impact the variable of interest throughout the historical period.

For example, the reconstructed skew surges were compared to the systematic ones (recorded ones). All historic surges appear to be almost at least as high as the highest systematic surge (almost equal to 1.30 m). The reconstructed skew surge heights obtained from the tide gauge data, the quantified surges from the literature and the reconstructed values from this study were also compared. Skew surges were plotted on the same graphic, as the hypothesis is made, that water levels measured at the tide gauge and the different locations of Dunkirk harbor are comparable.

On the other hand, we cannot conclude on the modification on harmonic constituents for the 19th century or the early 20th century because there are no high-frequency tide gauge observations in Dunkirk harbor before 1956. So we do not know to what extent work carried out on the channel (digging ...) and its modification and artificialization may have impacted the local hydrodynamics throughout time. Still, historic tide gauge data from Dunkirk is currently being digitalized and reconstructed at the French Oceanographic Service (SHOM - Service Hydrographique et Océanographique de la Marine) and University of Cote d'Opale: (Latapy et al., 2017) found approximately 10 years of high frequency data between 1865 and 1875. Once this data reconstructed, a detailed analysis of harmonic constituents will be performed, if the data quality is good enough.

Further It is worth noting that the current tide gauge is situated at the entrance of the harbor. The predicted water levels may differ within the inner harbor area, where the reconstructed surges were estimated. Hydrodynamic modelling could help estimate the difference between water levels at the entrance of the harbor area (Bulteau et al., 2015). |
| 3-if the results were considered as "final", and could thus be used as such for coastal management in the area of interest. I thought it was the case when I first read this paper. But the authors confirmed that more historical data have been collected and were not used here because it was the "subject of a second article". | *The reviewer: "… second article"*

The papers don't have the same objectives … The submission of the 2[nd] paper is still in progress and its content is still confidential…

The text talking about a future paper (in lines 445-446 of the 1st submitted version of the paper) were changed in the last version (Appendix 2 - §2) to say that work is ongoing.

" A 1740 sketch showing a profile of the Dunkirk harbor dock is presented in the lower panel of Fig. 3 for illustrative purposes only. The use of these plans and sketches in the estimation of some historical storm surges is ongoing."

The exploitation of these archives is very difficult. The additional information that will be presented from the second article is certainly important but I confirm that it adds nothing new and nothing more to our frequency estimates that are made in this article. As it is concluded in the present paper, we have achieved some stability and robustness in the results as the addition of new data will not have a significant impact. |

Authors propose to add a summary of all these points at the end of the same section "Data quality control". The following § was already written in section "Data Quality Control" of the current version of the article:

" Nevertheless, all types of data require quality control and need to be corrected and homogenized if necessary to ensure that the data are reflecting real and natural variations of the studied phenomena rather than the influence of other factors. This is particularly the case for historical data that have been taken in different site conditions and have not been taken using modern standards and techniques (Brázdil et al., 2010)."

It is replaced by the following two paragraphs:

1- at the end of section 3.3 Data quality control:

" Some other data quality related issues must be dealt with especially when using old data and when merge it with recent ones in a same inference: how to deal with old data uncertainties? How to deal with the evolution of some physiographic parameters around the site of interest (bathymetry, topography, land cover, etc.)? To what extent can we be sure that events which occurred hundreds of years ago are representative of the actual risk level?

All types of data require indeed quality control and need to be corrected and homogenized if necessary to ensure that they are reflecting real and natural variations of the studied phenomena rather than the influence of other factors. This is particularly the case for historical data that have been taken in different site conditions and have not been taken using modern standards and techniques (Brázdil et al., 2010). And finally, as mentioned in the introductory section, the use of old data improves significantly the frequency estimation of extreme events even they are inaccurate. The objective of the present paper is then to collect the information and to quantify it in order to obtain approximate values of the variable of interest, without seeking accurate reconstructions."

2- at the end of section 3.4 The historical surge dataset :

"It was concluded that all historic surges appear to be almost at least as high as the highest systematic surge. In response to the specific question: what could impact the variable of interest throughout the whole historical period? old and recent data and some physiographic conditions were then compared. For example, the reconstructed skew surges were compared to the systematic ones. The reconstructed skew surge heights obtained from the tide gauge data, the quantified surges from the literature and the reconstructed values from this study were also compared, as the hypothesis is made, that water levels measured at the tide gauge and the different locations of Dunkirk harbor are comparable. At this point we're not able to conclude on the evolution of the tides throughout the centuries. Historic tide gauge data from cities in the north of France is currently being digitized and reconstructed at the French Oceanographic Service (SHOM - Service Hydrographique et Océanographique de la Marine) and University of Cote d'Opale (Latapy et al., 2017). Further, it is worth noting that the current tide gauge is situated at the entrance of the harbor. The predicted water levels may differ within the inner harbor area, where the reconstructed surges were estimated. Hydrodynamic modelling could help estimate the difference between water levels at the entrance of the harbor area (Bulteau et al., 2015)."

The reference (Latapy et al., 2017) has been added in the references section.

Please see the the track changes version of the paper.

**Specific Comments**

| Comment | Response to reviewer |
|---|---|
| Specific comments
-Several syntax errors remain (e.g. "marine flooding"). | <li>According to us, "marine flooding" is not a syntax error.</li><li>A major revision of English for errors (syntax, grammar, spelling and vocabulary) was made by professionals (Technicis Translations).</li><li>Some syntax errors have been corrected: highwater – high water; hightide: high tide; marine floods: coastal floods.</li> |
| -it is unfortunate that the study justifying the use of the Dunkirk site is not published. The authors might at least indicate which agency made the study, and give some details on how it was done. | The study was made by the Institut de Radioprotection et de Sureté Nucléaire, in France (in collaboration with French historians). It is unfortunately still confidential. |

**Point-by-Point response / reviewer # 2**
Yasser Hamdi
**General comments**

| Comments | Responses to comments |
|---|---|
| line 296 As depicted in Fig. 2 (to the right) -> (to the left) | Ok. |
| In the discussion, some comments will be changed if you use a different cumulative empirical distribution estimator (Hazen in place of Weibull for example). Indicate what type of estimator you used. | Weibull plotting position rule was used herein ( $p_{emp} = i/(n+1)$ ).

• A sentence was added to the text (last § of section 4.3) ;
• Two references were also added to the references list. |
| line 499 "The RLs of interest had increased by only 10 to 20 cm. This is an important element of robustness. Indeed, adding or removing one or more extreme values from the dataset does not significantly affect the desired RLs." This comment is not general but specific to this database. | This is not general. Indeed, because of the exhaustiveness of the information used in the inference, any information you add or you remove from the dataset will not influence significantly the theoretical distribution. So it is specific to this database. |
| line 504 "The relative widths of CIs with no archival HI included are 1.5 times larger than those given by the full model. This means that the user of the developed model is more confident in the estimations when using the additional HI collected in the archives."
Could you precise the method that you use to calculate confidence intervals. | The delta method. It was mentioned in the last sentence of the 1st § - section 5 (Results & Discussion): " 
[revised manuscript text omitted]

---

## Author Response (AR3)

November 6, 2018

Re: Resubmission of manuscript "Analysis of the risk associated to coastal flooding hazards: A new historical extreme storm surges dataset for Dunkirk, France", nhess-2017-417

Copernicus Publications
Editorial Support

Dear Editor,

Thank you for accepting our manuscript "Analysis of the risk associated to coastal flooding hazards: A new historical extreme storm surges dataset for Dunkirk, France" with minor corrections. We also thank all the reviewers for giving us the opportunity to improve it.

All the changes in the revised manuscript were made using "track changes". Also point by point answers were introduced. We would be very happy, if the revised manuscript will now fulfill the high standards of the "Journal of Natural Hazards and Earth System Sciences" for publication.

Thank you again for consideration.

Yours sincerely,

Yasser Hamdi

**Point-by-Point response**

Yasser Hamdi

**Minor corrections:**

| Comments | Responses to comments |
|---|---|
| - The first item after the colon should not be capitalized | All the corrections are taken into account in the updated manuscript. |
| L103 : However, as | |
| L102-105 : please check font size | |
| L112 : skew surge : it is | |
| L199 : a depression | |
| L209 : while in 1954 | |
| L301 : even if they are inaccurate | |
| L384-385 : « In response to… » : sentence not clear. « Old » with capital letter | |
| L390 : we are not able | |